# Decreased brain connectivity in smoking contrasts with increased connectivity in drinking

Wei Cheng[1,2,3†], Edmund T Rolls[1,3,4†], Trevor W Robbins[5,6], Weikang Gong[1,7], Zhaowen Liu[8], Wujun Lv[9], Jingnan Du[1], Hongkai Wen[3], Liang Ma[10], Erin Burke Quinlan[11], Hugh Garavan[12,13], Eric Artiges[14], Dimitri Papadopoulos Orfanos[15], Michael N Smolka[16], Gunter Schumann[11], Keith Kendrick[17], Jianfeng Feng[1,2,3,18]*

[1]Institute of Science and Technology for Brain-inspired Intelligence, Fudan University, Shanghai, China; [2]Key Laboratory of Computational Neuroscience and Brain-Inspired Intelligence (Fudan University), Ministry of Education, Shanghai, China; [3]Department of Computer Science, University of Warwick, Coventry, United Kingdom; [4]Oxford Centre for Computational Neuroscience, Oxford, United Kingdom; [5]Behavioural and Clinical Neuroscience Institute, University of Cambridge, Cambridge, United Kingdom; [6]Department of Psychology, University of Cambridge, Cambridge, United Kingdom; [7]University of Chinese Academy of Sciences, Beijing, China; [8]School of Computer Science and Technology, Xidian University, Xi'an, China; [9]School of Mathematics, Shanghai University Finance and Economics, Shanghai, China; [10]Beijing Institute of Genomics, Chinese Academy of Sciences, Beijing, China; [11]Centre for Population Neuroscience and Stratified Medicine (PONS) and MRC-SGDP Centre, Institute of Psychiatry, Psychology and Neuroscience, King's College London, London, United Kingdom; [12]Department of Psychiatry, University of Vermont, Vermont, United States; [13]Department of Psychiatry Psychology, University of Vermont, Vermont, United States; [14]Institut National de la Santé et de la Recherche Médicale, INSERM Unit 1000 'Neuroimaging & Psychiatry', University Paris Sud – Paris Saclay, University Paris Descartes, Service Hospitalier Frédéric Joliot and GH Nord Essonne Psychiatry Department 91G16, Orsay, France; [15]NeuroSpin CEA, Université Paris-Saclay, Gif-sur-Yvette, France; [16]Department of Psychiatry and Neuroimaging Center, Technische Universität Dresden, Dresden, Germany; [17]Key Laboratory for Neuroinformation of the Ministry of Education, School of Life Science and Technology, Center for Information in Medicine, University of Electronic Science and Technology of China, Chengdu, China; [18]School of Mathematical Sciences and Centre for Computational Systems Biology, Fudan University, Shanghai, China

*For correspondence:
jianfeng64@gmail.com

†These authors contributed equally to this work

Competing interests: The authors declare that no competing interests exist.

**Abstract** In a group of 831 participants from the general population in the Human Connectome Project, smokers exhibited low overall functional connectivity, and more specifically of the lateral orbitofrontal cortex which is associated with non-reward mechanisms, the adjacent inferior frontal gyrus, and the precuneus. Participants who drank a high amount had overall increases in resting state functional connectivity, and specific increases in reward-related systems including the medial orbitofrontal cortex and the cingulate cortex. Increased impulsivity was found in smokers, associated with decreased functional connectivity of the non-reward-related lateral orbitofrontal cortex; and increased impulsivity was found in high amount drinkers, associated with increased

functional connectivity of the reward-related medial orbitofrontal cortex. The main findings were cross-validated in an independent longitudinal dataset with 1176 participants, IMAGEN. Further, the functional connectivities in 14-year-old non-smokers (and also in female low-drinkers) were related to who would smoke or drink at age 19. An implication is that these differences in brain functional connectivities play a role in smoking and drinking, together with other factors.

DOI: https://doi.org/10.7554/eLife.40765.001

## Introduction

Recent statistics have shown not only the widespread use of cigarettes and alcohol, but also the co-occurrence of these high risk behaviors. Although dopamine mechanisms in the basal ganglia have been implicated in addiction there is also considerable evidence for cortical, especially prefrontal cortical, involvement (*Ersche et al., 2011*; *Goldstein and Volkow, 2011*; *Ersche et al., 2012*; *Ersche et al., 2013*). For example, a voxel-based morphometry (VBM) study found that smokers exhibited reduced gray matter volumes and gray matter densities in frontal regions including pre-frontal cortex, orbitofrontal cortex, and anterior cingulate gyrus, as compared with non-smokers (*Wang et al., 2015*). Resting state functional connectivity differences involving the prefrontal cortex and insula have also been described in smokers (*Fedota and Stein, 2015*; *Yuan et al., 2016*; *Bi et al., 2017*; *Sutherland and Stein, 2018*). A functional magnetic resonance imaging (fMRI) study found that increased ventromedial prefrontal cortex (vmPFC) and anterior cingulate cortex (ACC) activation during 'relaxing' trials was correlated with high alcohol cue–induced and stress-induced craving in early recovering alcohol-dependent patients (*Seo et al., 2013*). These neuroimaging studies in substance use behaviors suggest that the prefrontal cortex plays a key role in such behaviors.

The high levels of comorbidity in using cigarettes and alcohol demonstrated by a wealth of epidemiological and genetic data (*Farrell et al., 2001*; *Agrawal and Lynskey, 2006*) makes it important to consider the neurobiological commonalities and differences associated with these two drugs, especially when they are consumed together. Thus, *King et al. (2010)* found that consumption of a moderately intoxicating oral dose of alcohol increased ratings of the desire to smoke, and moreover that alcohol amplified ventral striatum reactivity to appetitive cues associated with smoking in young individuals who tend to use cigarettes in the context of alcohol intoxication.

Most studies have used relatively small sample sizes and focused on brain regions rather than brain circuits identified with functional connectivity. Further, common and distinct connectivity features underlying the use of different substances such as nicotine and alcohol have not been fully explored with large numbers of participants (*Zhu et al., 2017*; *Sutherland and Stein, 2018*). In this study, we examined whole-brain functional connectivity (i.e. not limited by particular hypotheses and therefore unbiased) in groups using nicotine, or alcohol, or both, based on a large sample size (831 subjects from the Human Connectome Project (HCP) (and cross-validated by another 1176 subjects from the IMAGEN collaboration as described in the Materials and methods) with resting-state functional magnetic resonance imaging (fMRI) data. The aim was to identify shared and distinct patterns of different functional connectivity in smokers and drinkers in a sufficiently large sample to ensure sufficient statistical power to provide robust findings on associations between functional connectivity and smoking and drinking. Moreover, by including assessment of smoking and drinking in the same study, we aimed to provide robust evidence on whether there were differences in functional connectivity in smokers and drinkers. We also investigated aspects of substance use and impulsive behavior that might relate to the changes in neural connectivity, and that might provide information about the causation and aetiology of substance use.

## Results

### Decreased functional connectivity (FC) patterns for smoking

In the HCP participants, 273 links had significantly lower resting state functional connectivity in the smoking group compared to the non-smoking group (after FDR correction p<0.005). *Table 1* shows the strengths of the top 30 significantly different FC links between smokers and non-smokers. *Figure 1A* shows in the lower triangle matrix that functional connectivity across all FC links of the smoking group relative to the non-smoking group was lower overall (*Figure 1* and *Figure 1—figure*

**eLife digest** To understand why people become addicted to alcohol or smoking, it is important to look at how the brains of people who use these substances may be different than those who abstain. Many studies show that substance use activates the reward systems in the brain via a chemical called dopamine. Changes or differences in parts of the brain that control decision-making and restraint also have been implicated in substance use.

Functional magnetic resonance imaging (fMRI) is one tool scientists can use to explore such differences. It can measure how well different parts of the brain are communicating with each other by measuring their activity when a person is at rest. The patterns of activity reveal which parts of the brain are working closely together or have high functional connectivity and which parts are less well connected, or have low functional connectivity.

Cheng, Rolls et al. measured the functional connectivity between different parts of the brain in people who smoke and people who drink alcohol. Smokers had a low overall functional connectivity between brain regions. Specifically, they had weaker connections involving two brain regions that help people change or stop a behavior, the lateral orbitofrontal cortex and inferior frontal gyrus. These differences may make people more impulsive and less able to resist smoking. The stimulating effects of nicotine may enhance communication between different parts of the brain, so people also may use it to overcome some underlying communication deficits. Those who drink alcohol had high overall functional connectivity. Reward-related systems, including the medial orbitofrontal cortex and the cingulate cortex, were especially strongly connected. This may make them more sensitive to the rewarding aspects of drinking, or more impulsive.

To confirm their results, Cheng, Rolls et al. analyzed fMRI data from another study. These showed that the characteristic differences in brain connectivity were already present in 14-year olds who would go on to drink or smoke at age 19. This suggests that these functional connectivity differences in the brain make people more likely to smoke or drink.

DOI: https://doi.org/10.7554/eLife.40765.002

supplements *1*, *2*, *3* and *4*); and the upper triangle matrix shows which FC links were significantly lower in the smoking group ($p < 0.005$, FDR corrected). The links that are lower in the smoking group are shown in *Figure 2B*. Many (90) of the significantly different links involved the lateral orbitofrontal cortex (AAL2 areas OFClat and Frontal_Inf_Orb_2) and the adjacent inferior frontal gyrus (pars triangularis BA45 and pars opercularis BA44) (*Table 1*). These areas had lower functional connectivity with areas such as the hippocampus, temporal lobe, supramarginal gyrus, and insula (*Figure 2*). *Figure 2B* and *Table 1* show that in addition to these lower functional connectivities of the lateral orbitofrontal cortex and related inferior temporal gyrus areas, there was also lower functional connectivity for the middle and superior frontal gyri, mid- and superior temporal gyri, precuneus, hippocampus, and basal ganglia (caudate, putamen, and pallidum). *Table 1* shows, by contrast with the lateral orbitofrontal cortex, that there were only two significant medial orbitofrontal cortex areas (OFCmed) in this set of different functional connectivities in the smoking group.

## Increased functional connectivity patterns for high vs low drinking

In the HCP participants, the top 30 significantly different FC links between the low drinking and high drinking group are shown in *Table 1* (and *Figure 3* shows all 214 significantly different FC links after FDR correction with $p < 0.05$). All the significantly different links are higher in the high level of drinking group (HA, high amount) compared to the low level of drinking group (LA, low amount) (except one FC between OFClat_R and Angular_R).

*Figure 1B* shows in the lower triangle matrix that across all links (except one) the functional connectivity of the HA group relative to the LA group was higher overall; and the upper triangle matrix shows the FC links that were significantly higher in the HA drinking group than the LA group ($p < 0.05$, FDR corrected). Those links that are higher in the HA group than the LA group are shown in *Figure 3*, which show that, in addition to much higher functional connectivity involving the left medial orbitofrontal cortex, there was also higher functional connectivity for the cingulate cortex, supramarginal gyrus, globus pallidus, and pre-/post-central cortex. *Table 1* shows the top 30 links

**Table 1** Top 30 Functional Connectivity links with t and p values for Smoking and Drinking.
A negative t value indicates reduced FC relative to the control group.

| Smoking | | | | | | | |
| --- | --- | --- | --- | --- | --- | --- | --- |
| **Functional connectivity** | | **p value** | **t value** | **Functional connectivity** | | **p value** | **t value** |
| Frontal_Sup_2_L | OFCmed_R | 9.63E-06 | -4.464 | Frontal_Inf_Orb_2_L | Rolandic_Oper_R | 6.60E-05 | -4.019 |
| Frontal_Sup_2_L | Frontal_Mid_2_R | 1.23E-05 | -4.409 | Parietal_Sup_R | Precuneus_L | 6.74E-05 | -4.014 |
| Frontal_Sup_2_L | Frontal_Inf_Oper_R | 1.57E-05 | -4.354 | Frontal_Sup_2_L | Parietal_Sup_R | 7.11E-05 | -4.001 |
| Frontal_Inf_Orb_2_L | Temporal_Sup_R | 1.85E-05 | -4.317 | Frontal_Inf_Orb_2_L | Occipital_Mid_R | 7.60E-05 | -3.984 |
| Frontal_Inf_Orb_2_L | SupraMarginal_R | 2.02E-05 | -4.297 | Frontal_Sup_2_R | Frontal_Inf_Oper_R | 7.75E-05 | -3.980 |
| Frontal_Inf_Oper_R | Frontal_Inf_Tri_L | 2.26E-05 | -4.271 | Frontal_Inf_Orb_2_L | Insula_R | 7.95E-05 | -3.973 |
| Caudate_L | Pallidum_L | 2.42E-05 | -4.255 | Caudate_R | Putamen_R | 7.99E-05 | -3.972 |
| Caudate_L | Putamen_L | 2.87E-05 | -4.216 | Frontal_Mid_2_L | Frontal_Inf_Oper_R | 8.07E-05 | -3.970 |
| Hippocampus_L | Temporal_Sup_L | 3.37E-05 | -4.179 | Frontal_Inf_Orb_2_R | Parietal_Sup_L | 8.59E-05 | -3.955 |
| Frontal_Inf_Orb_2_L | Parietal_Sup_R | 3.78E-05 | -4.152 | Amygdala_R | Temporal_Sup_R | 8.94E-05 | -3.945 |
| Frontal_Med_Orb_R | Parietal_Sup_R | 3.86E-05 | -4.147 | Frontal_Inf_Oper_L | OFClat_R | 9.31E-05 | -3.935 |
| Frontal_Inf_Orb_2_L | Temporal_Sup_L | 4.59E-05 | -4.106 | Frontal_Inf_Orb_2_L | Rolandic_Oper_L | 9.37E-05 | -3.933 |
| Frontal_Inf_Tri_L | SupraMarginal_R | 4.90E-05 | -4.090 | Frontal_Inf_Tri_L | Precuneus_R | 9.54E-05 | -3.929 |
| OFCmed_R | Parietal_Sup_R | 5.11E-05 | -4.080 | Frontal_Inf_Oper_L | Cingulate_Ant_R | 1.01E-04 | -3.914 |
| Frontal_Inf_Orb_2_L | Amygdala_R | 5.41E-05 | -4.067 | Frontal_Sup_2_R | Frontal_Mid_2_R | 1.02E-04 | -3.912 |
| Drinking | | | | | | | |
| **Functional connectivity** | | **p value** | **t value** | **Functional connectivity** | | **p value** | **t value** |
| Precentral_L | OFCmed_L | 6.02E-04 | 3.445 | Cingulate_Ant_L | Postcentral_L | 3.42E-04 | 3.598 |
| Precentral_R | OFCmed_L | 2.41E-04 | 3.689 | Cingulate_Mid_L | Postcentral_L | 1.86E-04 | 3.756 |
| Rolandic_Oper_R | OFCmed_L | 1.28E-04 | 3.851 | OFCmed_L | Postcentral_R | 4.73E-04 | 3.511 |
| Olfactory_L | OFCmed_L | 4.35E-05 | 4.112 | Cingulate_Ant_L | Postcentral_R | 2.72E-04 | 3.658 |
| Rectus_R | OFCmed_L | 3.25E-04 | 3.611 | Cingulate_Mid_L | Postcentral_R | 1.94E-04 | 3.745 |
| Frontal_Mid_2_R | OFCmed_R | 6.10E-04 | 3.442 | Cingulate_Ant_L | SupraMarginal_R | 5.47E-04 | 3.472 |
| OFCmed_L | OFCpost_L | 4.25E-04 | 3.540 | OFCmed_L | Heschl_L | 1.36E-04 | 3.834 |
| Precentral_L | Cingulate_Ant_L | 4.52E-04 | 3.523 | Cingulate_Ant_L | Heschl_L | 1.55E-04 | 3.803 |
| Precentral_R | Cingulate_Ant_L | 2.50E-04 | 3.680 | Cingulate_Mid_L | Heschl_L | 5.02E-04 | 3.495 |
| Rolandic_Oper_L | Cingulate_Ant_L | 4.44E-04 | 3.528 | Cingulate_Mid_R | Heschl_L | 4.19E-04 | 3.544 |
| Rolandic_Oper_R | Cingulate_Ant_L | 1.45E-04 | 3.819 | SupraMarginal_R | Heschl_L | 2.68E-04 | 3.661 |
| Insula_R | Cingulate_Ant_L | 5.27E-04 | 3.481 | SupraMarginal_R | Heschl_R | 3.72E-04 | 3.575 |
| Rolandic_Oper_R | Cingulate_Ant_R | 5.22E-04 | 3.484 | Cingulate_Ant_L | Temporal_Sup_L | 1.49E-04 | 3.812 |
| OFCmed_L | Cingulate_Ant_R | 5.34E-04 | 3.478 | Cingulate_Ant_R | Temporal_Sup_L | 5.61E-04 | 3.464 |
| Precentral_R | Cingulate_Mid_L | 5.75E-04 | 3.458 | Cingulate_Mid_R | Temporal_Sup_L | 4.55E-04 | 3.521 |

DOI: https://doi.org/10.7554/eLife.40765.003

for the high drinking group, which are all higher compared to the low drinking (LA) group. Ten of the top 30 links involved higher functional connectivity of the medial orbitofrontal cortex, and 13 higher functional connectivity of the anterior cingulate cortex.

## Comparisons between drinking and smoking

For the HCP participants, *Figure 1C* shows the distribution of the t values of all 4371 FC links (i.e. between all 94 AAL2 areas) in the drinking and smoking groups respectively. Almost all links across the whole brain were increased in the drinking group but decreased in the smoking group. Further analysis shows that the t values of the links in the smoking group were significantly negatively correlated with those in the drinking group ($r = -0.289$, $p < 10^{-10}$) (*Figure 1D*) and this finding was well

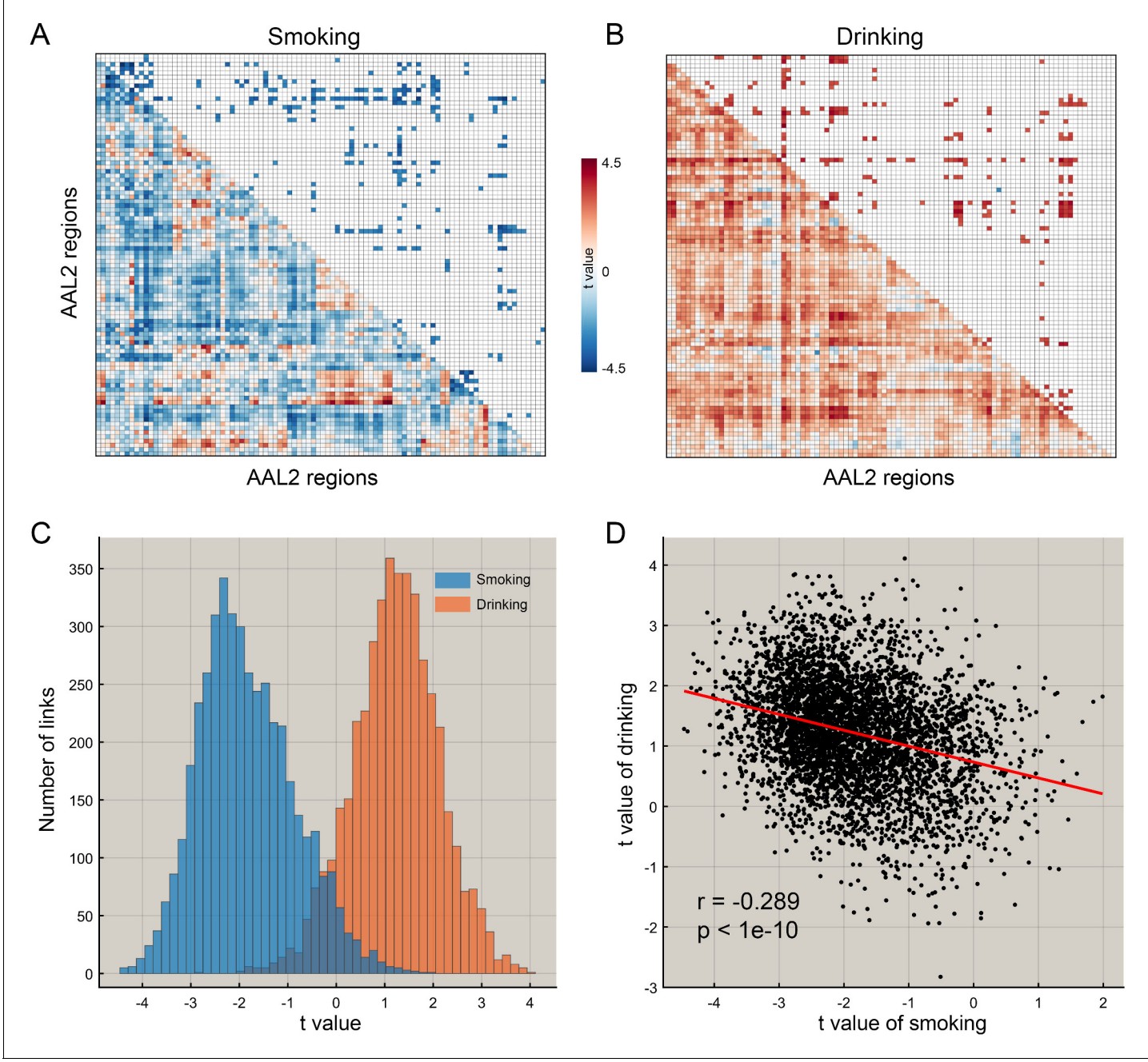

**Figure 1.** The difference of functional connectivity in smoking group and drinking group. (A) The matrix of t values for the smoking group. The lower triangle matrix shows the functional connectivities significantly associated with smoking. The baseline for the smoking was the never smoking group. The upper triangle matrix shows the significant links after multiple comparison correction (FDR corrected, p<0.005; no positive link survived correction for multiple comparisons). The regions are the AAL2 regions in the order shown in *Supplementary file 1*. (B) The matrix of t values for the drinking group. The lower triangle matrix shows the functional connectivities significantly associated with the amount of drinking per day. The baseline for the drinking was the low drinking group. The upper triangle matrix shows the significant links after multiple comparison correction using FDR correction (FDR corrected, p<0.05). The regions are the AAL2 regions in the order shown in *Supplementary file 1*. (C) The distribution of t values for the two groups for all AAL2 functional connectivity links. (D) The correlation between the t values of the above contrasts for the drinking and smoking groups. A positive t value indicates a stronger FC in a substance use group compared to the control group here and in the other Figures.

DOI: https://doi.org/10.7554/eLife.40765.004

The following figure supplements are available for figure 1:

**Figure supplement 1.** A comparison of the whole brain functional connectivity difference patterns between different pipelines of data preprocessing.

DOI: https://doi.org/10.7554/eLife.40765.005

*Figure 1 continued on next page*

*Figure 1 continued*

**Figure supplement 2.** Whole brain functional connectivity difference pattern using the fconn atlas (Shen *et al.,* 2013) for comparison with the AAL2 atlas results shown in *Figure 2*.
DOI: https://doi.org/10.7554/eLife.40765.006
**Figure supplement 3.** The correlation between four scans.
DOI: https://doi.org/10.7554/eLife.40765.007
**Figure supplement 4.** Comparison of whole brain functional connectivity difference patterns between male and female groups.
DOI: https://doi.org/10.7554/eLife.40765.008

cross-validated in an independent IMAGEN dataset (Figure 7C, r = −0.442, p<$10^{-10}$). This finding indicates a complementarity between the functional connectivities in drinkers and smokers. That is, if links were high in the smoking group, they tended to be low in the drinking group. This provides evidence that a difference in one direction (lower) may relate to smoking, and in an opposite direction (higher) to drinking.

We also performed a direct comparison between drinkers and smokers, to ensure that the results were not subject simply to possible baseline differences in the different control groups. In this contrast, we found, as predicted, that links involving the lateral orbitofrontal cortex (Frontal_Inf_Orb_2 and OFClat) were significantly (with FDR correction) positive, indicating that the lateral orbitofrontal cortex links are lower in smokers than drinkers. Further, also as predicted, medial orbitofrontal cortex and related areas (including OFCpost and Olf) were significantly more positive, indicating that the medial orbitofrontal cortex links are greater in drinkers than smokers, consistent with the other analyses in this investigation. Details are provided in the source data of *Figure 4*.

## Comparison between smokers, drinkers, and both smokers and drinkers

To analyse functional connectivity in those who both smoke and drink, the following comparison between four groups was made, with a common baseline group who neither smoked nor drank. Functional connectivity was compared over four groups (i.e smoking only n = 60, drinking only n = 219, and both smoking and drinking n = 143) using a common baseline comparison group (a no smoking and low drinking group n = 198) from the HCP dataset. *Figure 4A* shows the distribution of t values of all the links for all AAL2 areas for the three comparisons. This shows that the functional connectivity values for the smoking plus drinking group were similar to those for the only smoking group.

For *Figure 4B–D* only the significantly different links identified in the HCP dataset as shown in *Figures 2* and *3* are considered. *Figure 4B* confirms that all these links are lower in the smoking only group. *Figure 4C* confirms that almost all these links are higher in the drinking only group. *Figure 4D* shows that there are mainly lower links in the group that both smokes and drinks. *Figure 4—source data 1* shows that for the links that are lower in the smoking group as shown in *Table 1*, they are also lower in the smoking only group, and the both smoking and drinking group. *Figure 4—source data 2* shows that for the links that are higher in the drinking group as shown in *Table 1*, they are also higher in the drinking only group, while some are higher and some are lower in the both smoking and drinking group.

Thus the group that both smokes and drinks shows a combination of the links found to be different in the smoking group, and of those found to be different in the drinking group. However, it was noticeable that in the combined smoking and drinking group the difference in functional connectivity from the controls for the increases and decreases (expressed as a t value) found were smaller than in the only drinking or only smoking group (*Figure 4*).

In summary, participants who both smoked and drank had a combination of the specifically different links in those who only smoked or only drank, and these links had on average lower functional connectivity. An implication is that the differences in functional connectivity in smokers and drinkers described in this paper can, when combined, be related to, and potentially contribute to, both smoking and drinking in the same individual.

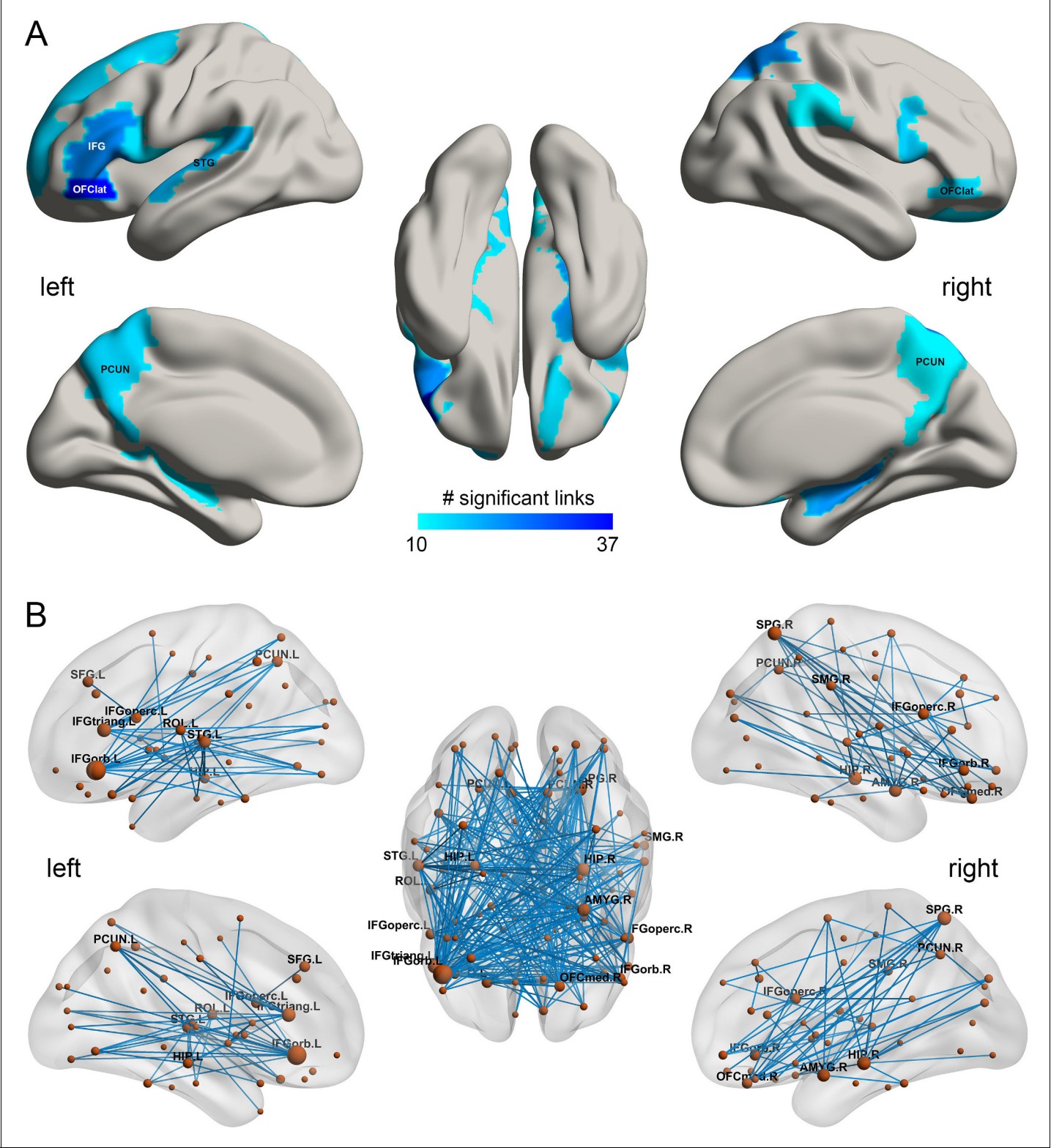

**Figure 2.** The smoking associated functional connectivities. (**A**) The AAL2 areas with different functional connectivity (measured by the number of significant links FDR p<0.005) involving that area, using a threshold of 10 links for the smoking compared to the non-smoking group. IFG – inferior frontal gyrus; OFClat –lateral orbitofrontal cortex; PCUN – precuneus; STG – superior temporal gyrus. (**B**) Significant links (FDR p<0.005) between the AAL2 areas with blue showing a decrease for the smoking group compared to the non-smoking group. The left hemisphere is on the left of each brain diagram in this and other Figures.

*Figure 2 continued on next page*

*Figure 2 continued*

DOI: https://doi.org/10.7554/eLife.40765.009

The following source data is available for figure 2:

**Source data 1.** All significant links (FDR p<0.05) between the AAL2 areas for the smoking group.

DOI: https://doi.org/10.7554/eLife.40765.010

## Correlation between the functional connectivity and the amount of smoking and drinking

*Figure 5A,B* shows that for the smoking group, there was a negative correlation between the mean strength of functional connectivity for both the significantly different links (n = 369, r = −0.102, p=0.031, permutation test) and the links for the whole brain (n = 830, r = −0.080, p=0.026) with the frequency of smoking across individuals. (This is consistent with the finding that the functional connectivity is lower in smokers.). It should be noted that all correlation analyses involving identified links were performed only within the smoking (or high drinking) group.

Figure 5C, D shows that for the drinking group, there was a positive correlation between the mean strength of FCs for the links for the whole brain (n = 782, r = 0.075, p=0.037) and a trend also for significantly different links (n = 471, r = 0.069, p=0.07, permutation test) and the amount of drinking across individuals (drinks per drinking day in the past 12 months). (This is consistent with the finding that the FCs are higher in drinkers).

Further, we found that this pattern of changes with the amount of use also applied to most of the significant links for both smoking (*Figure 5E*) and drinking (*Figure 5F*). Specifically, 269 out of 273 links associated with smoking showed a negative correlation with the frequency of smoking, and 54 of these 273 identified links were significant (p<0.05 uncorrected). In addition, 177 out of 214 links associated with drinking showed a positive correlation with the drinks per drinking day, and 38 of these 214 identified links were significant (p<0.05 uncorrected).

## Correlation between the functional connectivities involved in smoking and drinking, and impulsivity (assessed by temporal discounting)

We also investigated the relationship between the amount of smoking and drinking, with inhibitory control related to impulsivity (measured by the delay discounting score). *Figure 6A,B* shows that there was a negative correlation between the frequency of smoking and both the delay discounting scores, DDisc_AUC_200 (n = 830, r = −0.120, p=6.0 × $10^{-4}$) and DDisc_AUC_40K (n = 830, r = −0.088, p=0.012). There was also a negative correlation between the amount of drinking and both the delay discounting scores, DDisc_AUC_200 (n = 782, r = −0.122, p=7.0 × $10^{-4}$, *Figure 6C*) and DDisc_AUC_40K (n = 782, r = −0.033, p=0.036, *Figure 6D*). Thus higher impulsivity was associated with more drinking and more smoking in the drinking and smoking groups. Furthermore, canonical correlation analysis (CCA) revealed a significant model that relates functional connectivity levels to the delay discounting score for the drinking group (*Figure 6G*, n = 782, r = 0.575, p=9.9 × $10^{-4}$; *Figure 6H*, r = 0.572, p=0.002) with the same direction for the smoking group (*Figure 6E*, n = 830, r = 0.643, p=0.005; *Figure 6F*, r = 0.610, p=0.185). (The links involved in this analysis were all of the links shown in *Figure 2B* (for smoking) and *Figure 3B* (for drinking), 'all links' (with a different number of links for the two cases)).

The findings are consistent with decreases of functional connectivity in areas such as the lateral orbitofrontal cortex and the adjacent inferior frontal cortex being associated with greater impulsivity in the smoking group as measured with the delay discounting score.

For the drinking group, the significant increases in functional connectivity in the medial orbitofrontal cortex found in this group (*Figure 3B*) were associated with greater impulsivity. This may indicate that higher functional connectivity of the reward-related medial orbitofrontal cortex areas (*Rolls, 2017*; *Rolls, 2018b*) can be associated with increased impulsivity, possibly arising because rewards are driving behavior more strongly.

We further found that these measures of impulsivity had relatively high correlations, with respect to the 68 behavior measures in the HCP, with the smoking and drinking, and their related functional connectivities.

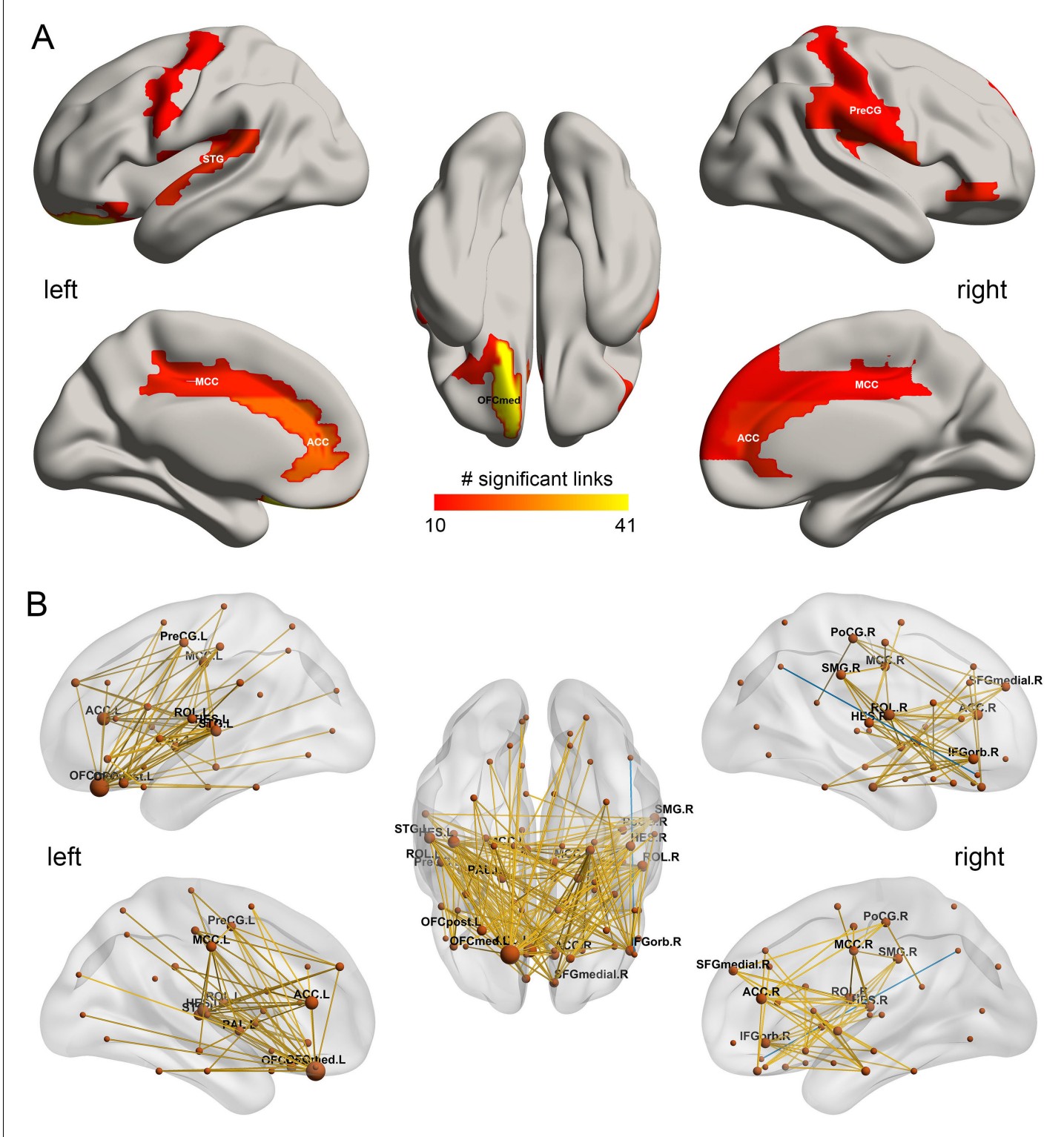

**Figure 3.** The drinking associated functional connectivities. (**A**) The AAL2 areas with different functional connectivity (measured by the number of significant links FDR p<0.05) involving that area, using a threshold of 10 links for high drinking compared to the low-drinking group. OFCmed – medial orbitofrontal cortex; ACC – anterior cingulate gyri; MCC – middle cingulate gyri; PreCG – precentral gyrus; STG – superior temporal gyrus; (**B**) Significant links (FDR p<0.05) between the AAL2 areas with red showing a increase for the drinking group.

DOI: https://doi.org/10.7554/eLife.40765.011

The following source data is available for figure 3:

*Figure 3 continued on next page*

*Figure 3 continued*

**Source data 1.** All significant links (FDR p<0.05) between the AAL2 areas for the drinking group.
DOI: https://doi.org/10.7554/eLife.40765.012

## Cross-validation using the IMAGEN dataset

Using a separate dataset (IMAGEN (*Schumann et al., 2010*)) we cross-validated the findings, as follows, with details provided in the Supplementary Material.

The results of this cross-validation for smoking are shown in *Figure 7*. *Figure 7A* shows the distribution of t values of the links between all AAL2 areas in the HCP and IMAGEN dataset at age 19. (Age 19 was used because by then some participants were smoking, and some were not.) The lower global (i.e. overall) functional connectivity found with the HCP dataset was confirmed in the IMAGEN dataset, with significant differences from zero at age 19 for smoking (t = −55.53, p<10$^{-10}$, one sample t-test). With respect to cross-validation of the specific links identified in the HCP dataset, *Figure 7D* shows that these links also had a significantly lower distribution of values compared to the comparison control group for the IMAGEN dataset at age 19 (one sample t-test, t = −13.00, p<10$^{-10}$).

For the drinking, the results of the cross-validation are also shown in *Figure 7*. *Figure 7B* shows the distribution of t values of the links between all AAL2 areas in the HCP and IMAGEN datasets at age 19. (Age 19 was used because by then some participants were drinking, and some were not.) The higher global functional connectivity found with the HCP dataset was confirmed in the IMAGEN dataset, with a significant difference from zero for the IMAGEN dataset at age 19 (t = 98.00, p<10$^{-10}$, one sample t-test). *Figure 7E* shows that the significant links for drinking identified in the HCP dataset also had a significantly higher distribution of t values compared to the comparison control group for the IMAGEN dataset at age 19 (one sample t-test, t = 15.32, p<10$^{-10}$). As described in the Supplementary Material, the comparison group was the low-drinkers.

## Association between the functional connectivity at age 14 and the smoking and drinking at age 19

To investigate whether the functional connectivity at 14 might be related to whether a participant became a smoker by age 19, a longitudinal analysis with the same IMAGEN participants was performed. The 19 years olds were split into two approximately equal size groups, of smokers vs non-smokers. For each link, its value in the 14 years olds was compared between those who were smokers and non-smokers at age 19 with a two sample t test. For the overall connectivity, this provided 4371 t values. A one-sample t-test was then performed for whether the mean of these t-values was lower than 0 in those who became smokers at 19. That test was significant (t = −47.51, p<10$^{-10}$, *Figure 7A*), providing evidence that the value overall of the links was lower at 14 in those who became smokers at 19. With respect to the significantly different links between smokers and non-smokers identified in the HCP dataset, the test was also significant (t = −12.23, p<10$^{-10}$, *Figure 7D*), providing evidence that the values of the significantly different links found in the HCP dataset were lower at age 14 in the IMAGEN dataset in those who became smokers at 19. An implication is that the low functional connectivities may not simply be produced by smoking, but may instead contribute to the tendency to smoke.

Similar tests for drinking found analogous effects (t = 15.31, p<10$^{-10}$ for the overall links (*Figure 7B*), and t = 7.06, p<10$^{-10}$ for the specific links (*Figure 7E*)) in female, but not in male, IMAGEN participants.

## The relation between impulsivity at age 14 and the smoking and drinking at age 19

The analyses of impulsivity with the IMAGEN dataset are considered next. First, the impulsivity is significantly positively correlated with the frequency of smoking (r = 0.106, p=1.4 × 10$^{-4}$) and the amount of drinking (r = 0.064, p=0.022) at age 19. The results for impulsivity at age 14 are shown in *Figure 7F*. The 14 year olds who will smoke at age 19 (n = 421) have higher impulsivity than those who will not smoke at 19 (n = 184) (t = −3.72, p=2.20 × 10$^{-4}$). The 14 year olds who will drink at

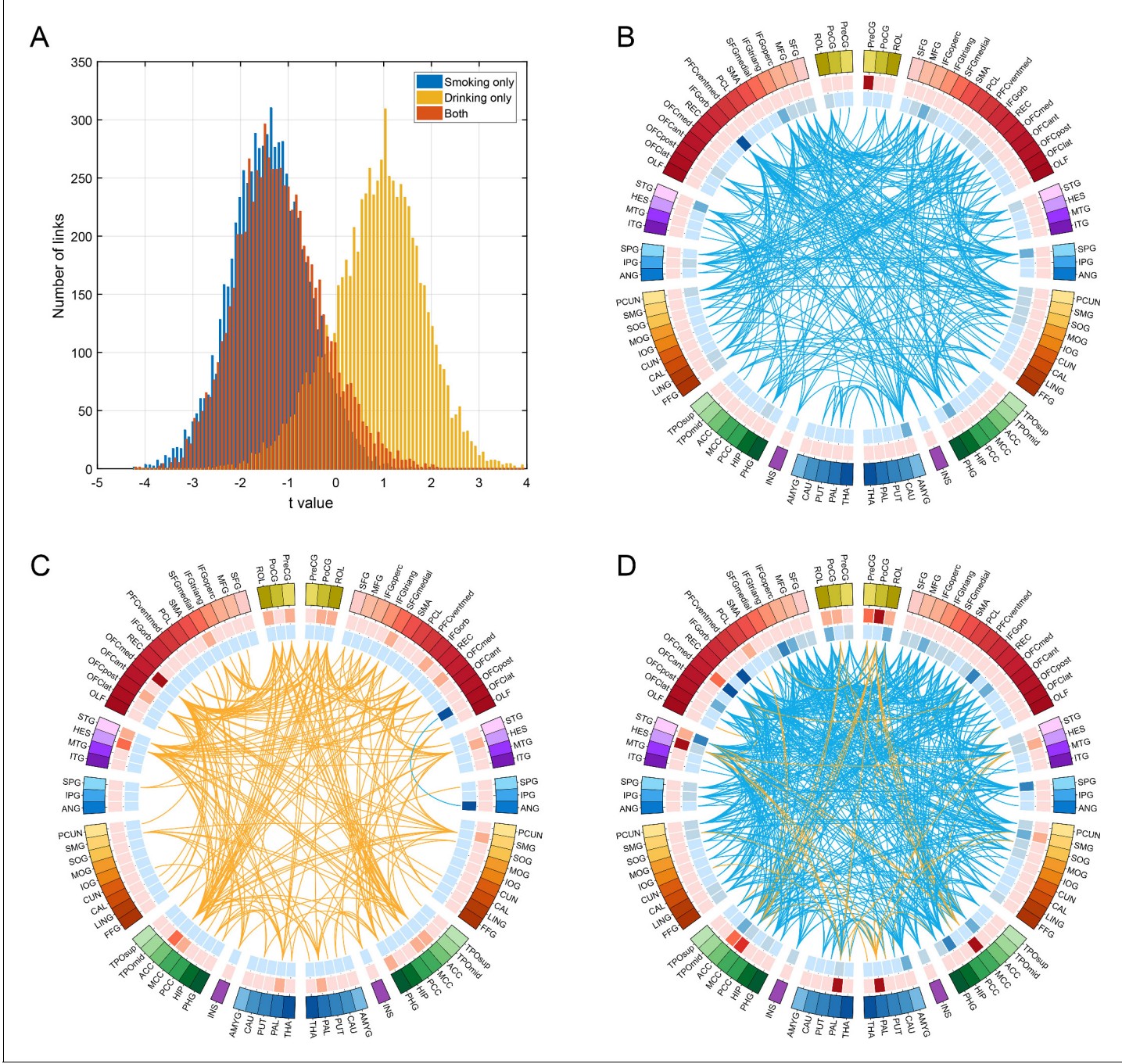

**Figure 4.** Comparison of functional connectivity in three groups (i.e smoking only n = 60, drinking only n = 219, and both smoking and drinking n = 143) using a common baseline comparison group (a no smoking and low drinking group n = 198) from the HCP dataset. (**A**) shows the distribution of t values of all the links for all AAL2 areas for the three comparisons. A positive t value indicates a stronger FC in a substance use group compared to the control group. For B-D, only the significantly different links identified in the HCP dataset as shown in *Figure 3* are considered. (**B**) shows that all these links are decreased in the smoking only group. (**C**) shows that all these links are higher in the drinking only group. (**D**) shows that there are both higher and lower links in the group that both smokes and drinks.

DOI: https://doi.org/10.7554/eLife.40765.013

The following source data is available for figure 4:

**Source data 1.** For the FC links identified as involved in smoking: comparison of functional connectivity between three groups: smoking only; drinking only; and both smoking and drinking.

DOI: https://doi.org/10.7554/eLife.40765.014

*Figure 4 continued on next page*

*Figure 4 continued*

**Source data 2.** For the FC links identified as involved in drinking: comparison of functional connectivity between three groups: smoking only; drinking only; and both smoking and drinking.
DOI: https://doi.org/10.7554/eLife.40765.015

age 19 (n = 527) have higher impulsivity than those who will not drink at 19 (n = 298) (t = −3.67, p=2.60 × 10$^{-4}$). Thus higher impulsivity at 14 is associated with who will smoke, or drink, at age 19.

## Discussion

Smokers and drinkers had contrasting patterns of cortical functional connectivity involving the orbitofrontal cortex. Smokers relative to non-smokers exhibited reduced functional connectivity involving the lateral orbitofrontal cortex which is implicated in stopping and changing behavior and thus to

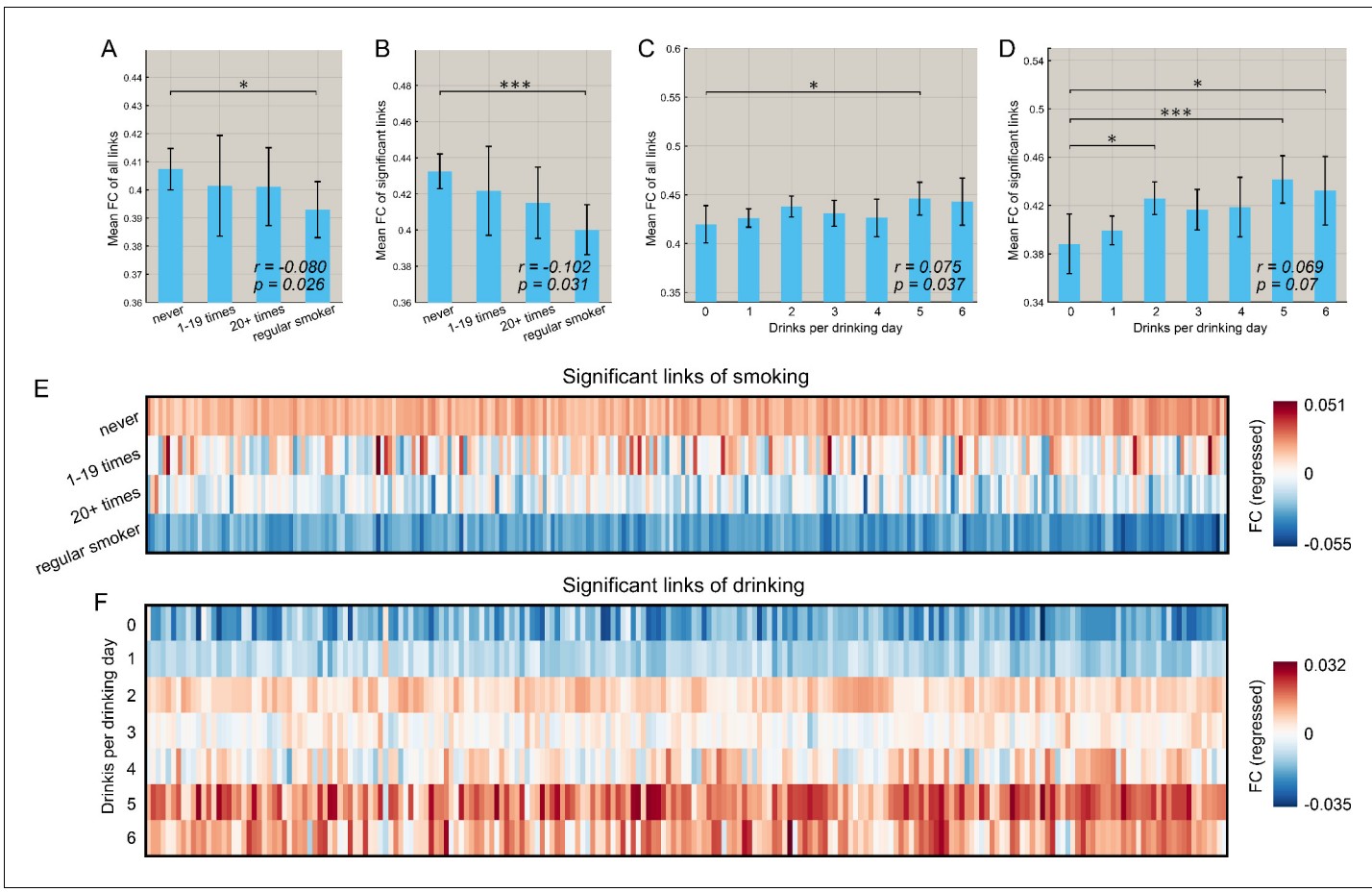

**Figure 5.** Correlation between the strength of functional connectivity and the amount of substance use behavior. (A) Comparison of the mean strength of all functional connectivities in subgroups that smoked for different numbers of times. The error bars show the standard error of the mean (SEM) in this and other Figures. The mean strength of all functional connectivities in the regular smoking group is significantly lower than in the non-smoking group (p=0.026). The correlation between the strength of FC links and the amount of substance use behavior are shown in the lower right corner (same for other figures). (B) Comparison of the mean strength of significantly different functional connectivities in subgroups who had smoked for different numbers of times. The mean strength of significantly different functional connectivities of the regular smoking group is significantly lower than for the non-smoking group (p=0.031). (C) Comparison of the mean strength of all functional connectivities in subgroups with different amounts of drinking. (D) Comparison of the mean strength of significantly different functional connectivities in subgroups with different amount of drinking. The relation between the strength of the different FCs and the smoking history (E) and the amount of drinking (F). The FCs are those that are significantly different between the substance use and control groups as shown in *Figures 2* and *3*. * indicates p<0.05, **p<0.005 and ***p<0.0005.
DOI: https://doi.org/10.7554/eLife.40765.016

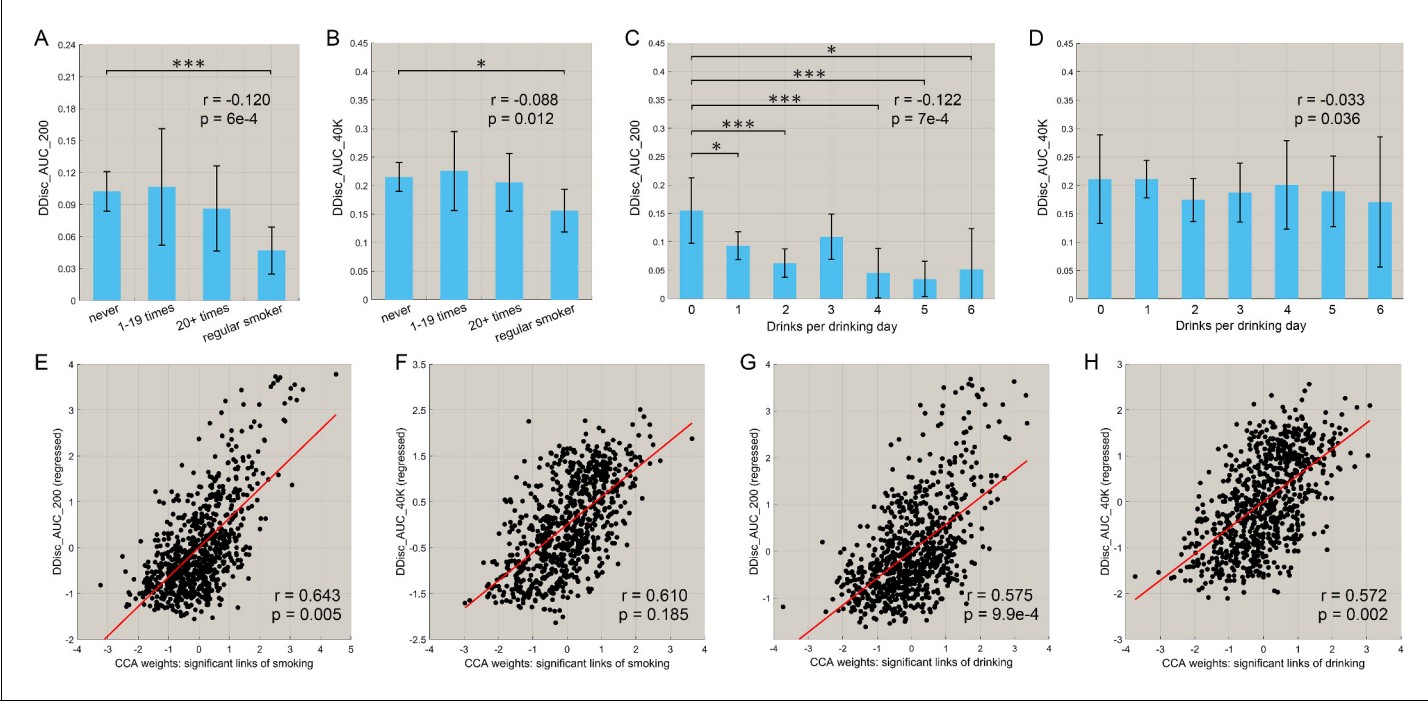

**Figure 6.** Correlation between the strength of functional connectivity and temporal discounting behavior (impulsivity). (A, B) The delay discounting score for different subgroups with different frequencies of smoking. A low score reflects choices for immediate reward as compared with longer term reward. The delay discounting scores are significantly lower in the regular smoking group compared with the non-smoker group. The correlations between the delay discounting scores and the amount of substance use behavior are shown in the top right corner (with the same for the other figures). The error bars show the standard error of the mean (SEM) in this and other Figures. (C, D) The delay discounting scores for subgroups with different amounts of drinking. The delay discounting scores are significantly lower in the high drinking group compared with the low drinking group. (E, F) The canonical correlation between the delay discounting score and the strength of the significantly different links in the smoking dataset. The scatter plot shows the delay discounting score versus the functional connectivity weights, with one point per participant. The high correlation shown here indicates a significant co-variation between the FCs and the delay discounting score. (G, H) The same for the drinking group. (* indicates p<0.05, **p<0.005 and ***p<0.0005).

DOI: https://doi.org/10.7554/eLife.40765.017

impulsive-compulsive disorders, and in related areas including the inferior frontal gyrus, superior temporal gyrus, precuneus, and supramarginal gyrus (*Figure 2A*). By contrast, high drinkers relative to low drinkers had higher functional connectivity in networks including the medial orbitofrontal cortex (bilaterally though more on the *left*) which is involved in reward processing, and in related structures including the anterior cingulate cortex, parahippocampal cortex, supramarginal gyrus, insula, and superior temporal gyrus. In addition, the mean functional connectivity across the whole brain was *lower* in smokers than non-smokers. In contrast, mean functional connectivity across the whole brain was *higher* in high drinkers than low drinkers. The differences in both the overall and the identified links were correlated with the amount of smoking or drinking. For both the smoking and drinking, the differences in overall functional connectivity were correlated with increased impulsivity. These findings with the HCP dataset were cross-validated with the IMAGEN dataset. In addition, analysis of the IMAGEN dataset showed that for the significantly different links between smokers and non-smokers identified in the HCP dataset, those links were lower at age 14 in the IMAGEN dataset in those who became smokers at 19 (*Figure 7D*). An implication is that the low functional connectivities may not simply be produced by smoking, but may instead contribute to the tendency to smoke. The same was found for the global functional connectivity (*Figure 7A*). For drinking, analysis of the IMAGEN dataset showed that for the significantly different links between high drinkers and low drinkers identified in the HCP dataset, those links were higher at age 14 in the IMAGEN dataset in those who became high drinkers at 19 (*Figure 7B,E*) for females, though that effect was not significant in the male participants.

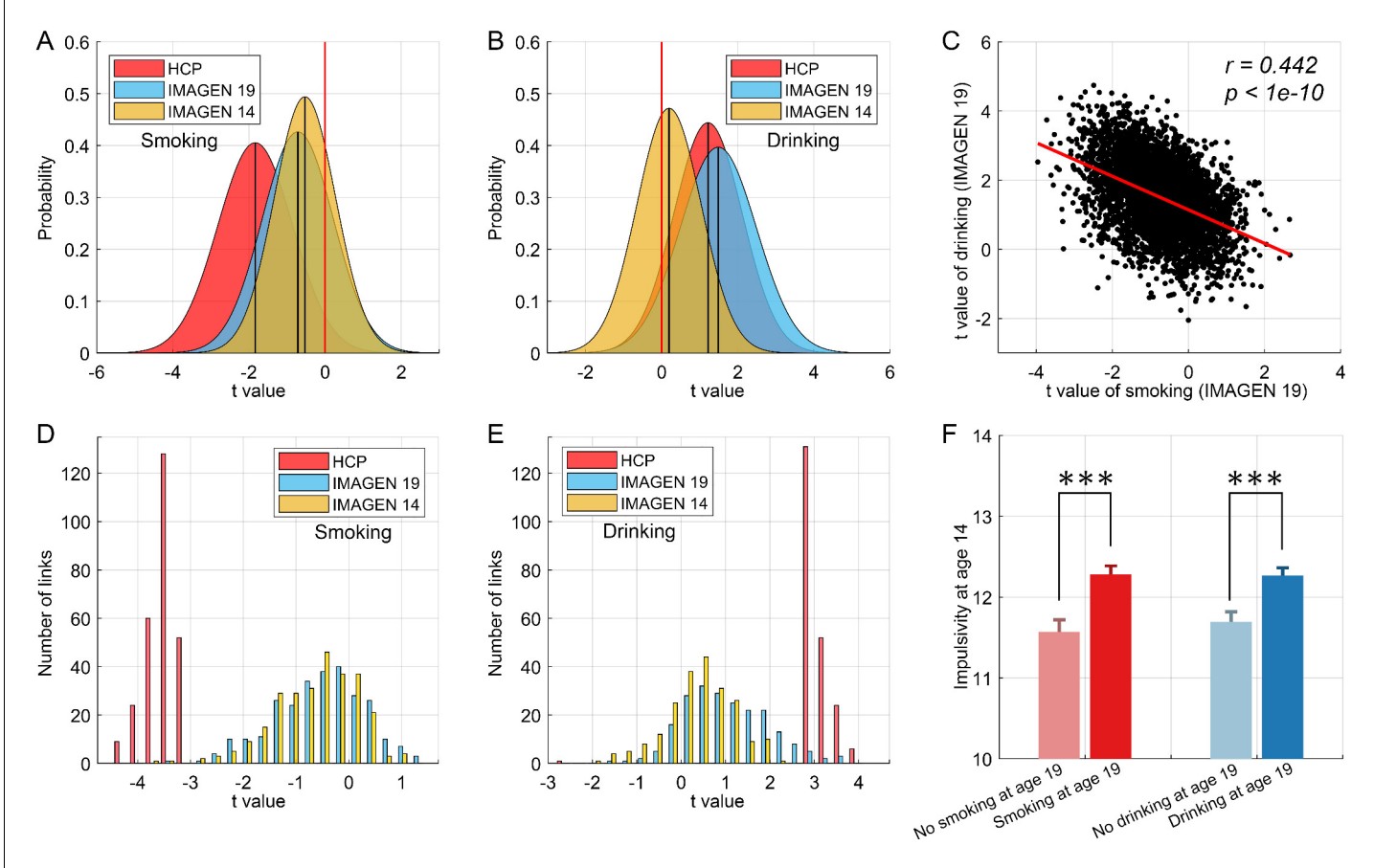

**Figure 7.** Cross-validation and causality analysis using the IMAGEN dataset. (**A**) The distribution of t values of all brain links in the three datasets (i.e. HCP, and IMAGEN at age 14 and 19) for smoking. (**B**) The same for the drinking group. The overall change pattern identified in the HCP dataset, that is decreased FC in smoking and increased FC in drinking, is consistent in the IMAGEN dataset. This is useful cross-validation. (**C**) The correlation between the t values of the contrasts for the drinking and smoking groups involving all links in the brain. (**D**) The distribution of t values of the significant links identified in the HCP dataset for all three datasets (i.e. HCP, IMAGEN at age 14 and 19) for smoking. (**E**) The same for the drinking group. (**F**) The results for impulsivity at age 14 from the IMAGEN dataset. This shows that the 14 year olds who will smoke at age 19 (n = 421) have higher impulsivity than those who will not smoke at 19 (n = 184) (t = −3.72, p=2.20 × 10^{−4}). The 14 year olds who will drink at age 19 (n = 527) have higher impulsivity than those who will not drink at 19 (n = 298) (t = −3.67, p=2.60 × 10^{−4}). ***p<0.001.
DOI: https://doi.org/10.7554/eLife.40765.018

The low functional connectivity in smokers relative to non-smokers especially involved the lateral orbitofrontal cortex (Brodmann area 47/12) which is implicated in stopping and changing behavior, including when reward is not received (*Kringelbach and Rolls, 2003*; *Robbins, 2007*; *Grabenhorst and Rolls, 2011*; *Rolls, 2014*; *Rolls, 2016*), and when there is an instruction to stop in the stop-signal task (*Deng et al., 2017*). This reduced connectivity also involved the inferior frontal gyrus, pars opercularis (BA 44) and pars triangularis (BA45). Further, damage to the lateral orbito-frontal cortex (*Berlin et al., 2004*; *Rolls, 2014*; *Rolls, 2016*) and to the right inferior frontal gyrus (*Aron et al., 2014*) produces impulsive behavior (measured for example in the stop-signal task), where impulsive behavior can be considered as behavior not directly inhibited by non-reward or punishment feedback signals (*Rolls, 2014*; *Rolls, 2017*). Consistent with this, lower functional connectivity involving the lateral orbitofrontal cortex in the present study was related to increased impulsiveness as measured by the delay discounting score (*Figure 6*). An implication is that smokers may smoke in part because of increased impulsiveness and reduced sensitivity to non-reward and punishment associated with reduced functional connectivity of the lateral orbitofrontal cortex and adjacent inferior temporal gyrus. Consistently, we found that the delay discounting score (DDS) was

positively correlated with the strength of many of the lateral orbitofrontal cortex links (in the HCP dataset). Thus high FC of the lateral orbitofrontal cortex (OFC) is correlated with low impulsivity.

The increased functional connectivity in drinkers relative to non-drinkers involved the medial orbitofrontal cortex which is involved in reward processing (*Grabenhorst and Rolls, 2011*; *Rolls, 2014*; *Rolls, 2017*) and structures to which it is connected including the anterior cingulate cortex, parahippocampal cortex, supramarginal gyrus, and superior temporal gyrus. We found that the delay discounting score (DDS) was negatively correlated with the strength of many of the medial orbitofrontal cortex links (in the HCP dataset). Thus high FC of the many medial OFC links was correlated with high impulsivity, which correlated in turn with the amount of drinking. In rats, steeper delay discounting (interpreted as greater impulsivity) is produced by lesions of the lateral OFC, and reduced discounting by lesions of the medial OFC (*Mar et al., 2011*), possibly homologous regions to those in the primate brain, suggesting a similar opponency of control over impulsivity by these prefrontal cortex regions to that found in the present investigation in humans.

In the short term just after administration, alcohol increases impulsivity and decreases behavioral inhibition (*Smith et al., 2014*) (and increases neurophysiological inhibition in some brain areas) (*Stephens et al., 2017*). The mechanism for the increase in impulsivity produced by alcohol may include reduced activity in the lateral orbitofrontal cortex, which tends to have activations reciprocally related to those of the medial orbitofrontal cortex (*O'Doherty et al., 2001*; *Kringelbach and Rolls, 2003*; *Rolls, 2016*; *Rolls et al., 2018a*). The fact that both drinkers and smokers exhibited greater impulsive choice is consistent with a possible causal role for impulsivity in either the initiation of drinking or smoking or else its exacerbation, as well as other behaviors associated with impulsivity, including aggression in the case of alcohol consumption (*Garofalo and Wright, 2017*).

To check the main findings of this investigation, and to control for possible differences in the comparison groups, we directly compared functional connectivity in drinkers and smokers. The hypotheses were that functional connectivity involving the lateral orbitofrontal cortex would be smaller in smokers than drinkers; that functional connectivity involving the medial orbitofrontal cortex would be greater in drinkers than in smokers; and that the mean functional connectivity would be higher in drinkers than non-smokers. All these hypotheses were confirmed by the direct comparison of drinkers – smokers in the Results section (see *Figure 4*).

Using a separate dataset (IMAGEN (*Schumann et al., 2010*)) we cross-validated the findings obtained with the HCP dataset. In addition, analysis of the IMAGEN dataset showed that there was an association between the low connectivity in smoking at age 14 (when participants were not smoking) and whether at age 19 individuals were smoking. This association was found for both the global functional connectivities (*Figure 7A,B*) and the specifically different functional connectivities (*Figure 7D,E*). A similar relation was found in drinkers, namely that there was an association between the high functional connectivitities at 14 and wherher an individual would be in the high drinking group at age 19. This was significant for females, but not for males. An implication is that the differences in brain functional connectivities identified in this investigation may play a causal role in who will become smokers or drinkers. It will be of interest in future to assess further whether the differences in functional connectivity described here in smokers and drinkers cause the smoking or drinking, for example in research on alcohol drinking endophenotypes or smoking endophenotypes (*Ray et al., 2010*; *Ducci and Goldman, 2012*). It is of course also a possibility that drinking may by increasing impulsivity exacerbate the drinking.

The low functional connectivity that we describe here in smokers in frontal and associated areas may be reflected in the hypofrontality that has previously been described in smokers and has been associated in genome-wide association studies with single-nucleotide polymorphisms (SNPs) in the human CHRNA5 gene, encoding the α5 nAcetylcholine receptor subunit, that increase the risks for both smoking and schizophrenia (*Barch et al., 2001*; *Hong et al., 2010*; *Tobacco and Tobacco and Genetics Consortium, 2010*); Schizophrenia Working Group of the *Schizophrenia Working Group of the Psychiatric Genomics Consortium, 2014*; *Koukouli et al., 2017*). So it is possible that smokers self-administer nicotine which acting on this cholinergic receptor may increase cortical neuronal firing, which stabilizes the activity of prefrontal attractor networks, making them more efficient at cognitive control tasks involved in executive function. This has been proposed for the smoking in schizophrenia (*Rolls et al., 2008*). Thus, we suggest that lower initial whole brain FC in smokers may encourage them to enhance their functional connectivity and hence the functioning of the neural

circuitry implicated in cognitive control; smoking can then be seen as a form of 'self-medication' which optimizes their behavior.

For female drinkers there was also an association between high functional connectivity at age 14 and who would be in the high drinking group at age 19. In this case a possible account for the drinking is that it is related to increased functional connectivity or sensitivity of medial orbitofrontal cortex and related reward circuitry, which makes the alcohol more rewarding, and which may also because of the increased sensitivity to reward produce some increase in reward-related impulsivity (*Whelan et al., 2012*). The acute effects of the alcohol may themselves reduce behavioral inhibition and increase the drinking, thus exacerbating the drinking (*Whelan et al., 2014*).

*Figures 1D* and *7C* show a negative correlation between the t values for all the connectivities in the brain for drinkers vs smokers. This may be related to the fact that high values of t for the functional connectivity relate to high drinking, and low values t for the functional connectivity relate to smoking. Thus the negative correlation reflects whether individuals are more likely to perform an action of either smoking or drinking. This reflects the fact that in this dataset those who smoke also are more likely to drink. A common underlying factor might be impulsivity.

We note that although there were overall higher in FC in drinkers, and lower in smokers, to place this in context, FCs were also smaller in males than females in this dataset in smokers and drinkers (*Figure 1—figure supplement 4*). Accordingly, whilst we would not wish to over-interpret the differences in global mean FC in drinking and smoking, the negative correlation between all FCs in drinking and smoking shown in *Figure 1E* is very suggestive that there is indeed complementarity of the differences found in drinkers and smokers. An interpretation is that smoking and drinking are related to opposite differences in some of many of the links involved. Examples from this investigation are that medial orbitofrontal cortex links are decreased in smokers, but increased in drinkers; and vice versa for lateral orbitofrontal cortex links.

The differences in functional connectivity described in this investigation were not due to any obvious possible confounding factors such as gender, ethnicity, level of education, marijuana use, or basic brain structure (volume of gray and white matter), which were regressed out of the analysis.

The use of resting state fMRI is useful for this type of investigation as it does not require task performance that might differ between participants, and such studies can for example predict behavior (*Rosenberg et al., 2016*) and task performance (*Tavor et al., 2016*) in different individuals. Structure based morphological comparisons for drinkers and smokers have been performed, but do not necessarily relate closely to what is found with functional investigations, as parts of the brain may have different physiology without much gross anatomical difference.

Possible limitations of this investigation are that linear analyses were used. It is also important to stress that all participants were healthy adults, and that the levels of smoking and drinking exhibited were not directly associated with diagnoses of substance dependence, although they may be relevant to these disorders. Thus the effect size of this study involving the normal use of alcohol and cigarettes could be lower than in a case/control study. But the design of this study, use of individuals in the general population, enable this investigation to have impact on public health, since the use of substances for most of the smokers and drinkers was within a normal range, and we could identify these changes by the use of two large-scale datasets. Although the current study was based on the general population, the findings are in general consistent with previous studies based on clinical populations. For example, chronic nicotine exposure associated with lower functional connectivity was reported in a number of studies (*Hong et al., 2009*; *Weiland et al., 2015*). In addition, many studies report that the orbitofrontal cortex, the key region identified in current study, play an important role in the neurobiological mechanisms of addiction (*Lucantonio et al., 2012*; *Hu et al., 2015*; *Koob and Volkow, 2016*). However, the results described here were obtained on populations from the general community, and do not necessarily apply to heavy smokers or drinkers. Moreover, we demonstrate with the IMAGEN dataset that the effect size is sufficient that there is a relation between the brain functional connectivities that we identify at age 14 and who will smoke at age 19. This is good evidence that the effects described in this paper are meaningful, in that the functional connectivity at age 14 can be related to what will happen in terms of smoking and drinking five years later. Additionally, we have removed several possible confounders that may distinguish the two groups, but it might be useful in future investigations to investigate the relation of depression to smoking and drinking. Another possible limitation is that impulsivity was assessed by delay discounting, and it would be of interest to examine the relation to other measures of impulsivity. Another

possible area in which future research might extend the findings reported here is that we focused on the amount of drinking using the measure 'drinking amount per drinking day', as the relation between the frequency of drinking and the functional connectivity was complex. But this enables us to make an interesting point. The amount of alcohol consumed on a drinking day, the 'amount' measure, may well be related to how rewarding the consumption of the alcohol is. Once started on a drinking day, an individual in the High Amount drinking group tends to have more, and this is consistent with the reward value being high. That in turn is consistent with the higher functional connectivity of the medial orbitofrontal cortex found in the High Amount drinkers in this investigation, and with the evidence that the medial orbitofrontal cortex contains a representation of reward value (*Rolls, 2014*; *Rolls, 2017*; *Rolls, 2018b*). Another possible limitation of the drinking analysis is that the control group was a low drinking group rather than a non-drinker group. The reason for this choice was that only 55 participants did not drink any alcohol in the past 12 months, which would have resulted in too small a sample size.

As shown in *Figure 4*, most of the links were lower in the group that both smokes and drinks. One explanation is that the effect size for the comparison between non-smokers and regular smokers is higher than the effect size for the comparison between low-drinkers and high drinkers.

In conclusion, we have related for the first time reduced functional connectivity involving the lateral orbitofrontal cortex and related inferior frontal gyrus to smoking. We interpret the smoking as being related to the increased impulsivity and decreased behavioral inhibition. In addition, smokers have globally lower functional connectivity. A possible interpretation is that smokers may self-administer nicotine in order to increase brain functional connectivity and thereby attention and alertness. We also describe for the first time that drinkers have increased functional connectivity of the medial orbitofrontal cortex, a reward area, and globally increased functional connectivity as well as impulsive behavior. An interpretation is that there is a role in drinking of increased reward sensitivity implemented in the medial orbitofrontal cortex. The increased impulsivity in drinkers may also be related to increased reward processing in the medial orbitofrontal cortex and its connected areas. These differences in functional connectivity may play a causal role, as shown by analyses showing that there was an association between smoking or drinking at age 19 and the functional connectivities at age 14 of individuals who at 14 are not smoking or drinking. In both smokers and drinkers, the increased impulsivity may exacerbate substance use. These findings open the way for further investigations of the extent to which these differences contribute to the substance use, or are caused by it. In any case, we note that there may be many factors that influence smoking and drinking, though we have revealed some of the possible underlying factors and brain mechanisms in this investigation.

## Materials and methods

### Participants and data preprocessing

The dataset used for this investigation was selected from the November 2015 public data release from the Human Connectome Project (HCP, N = 900), WU-Minn Consortium. Our sample includes 831 subjects (ages 22–35 years, 463 females) scanned on a 3 T Siemens connectome-Skyra scanner. The HCP consortium is a public shared large-scale neuroimaging dataset which aims to map macroscopic human brain circuits and their relationship to behavior in a large population of healthy adults (*Van Essen et al., 2012*; *Van Essen et al., 2013*) and which has been widely used in neuroimaging studies (*Finn et al., 2015*; *Smith et al., 2015*; *Glasser et al., 2016*). The sibships with individual having severe neurodevelopmental disorders, neuropsychiatric disorders or neurologic disorders were excluded, but individuals who are smokers, are overweight, or have a history of heavy drinking or recreational drug use without having experienced severe symptoms were included in HCP consortium. The details on the inclusion and exclusion criteria of HCP consortium were provided in the previous studies (*Van Essen et al., 2012*; *Van Essen et al., 2013*) and the HCP website (https://www.humanconnectome.org/). It should be noted that the HCP dataset was not specifically designed for the study of substance use, but it is useful for investigating the impact of alcohol and cigarette use in a community sample. Two resting state fMRI acquisitions on different days were used. The four resting-state runs of approximately 15 min each were acquired in separate sessions on two different days, with eyes open with relaxed fixation on a projected bright cross-hair on a dark background.

The WU-Minn HCP Consortium obtained full informed consent from all participants, and research procedures and ethical guidelines were followed in accordance with the Washington University Institutional Review Boards (IRB #201204036; Title: 'Mapping the Human Connectome: Structure, Function, and Heritability'). The demographic characteristics of participants are summarized in *Table 2*. More details of subjects are provided in the Supplementary Material and the collection and preprocessing of the data are provided at the HCP website (http://www.humanconnectome.org/).

We obtained minimally preprocessed R-fMRI data conducted using the HCP Functional Pipeline v2.0 (*Glasser et al., 2013*) involving gradient distortion correction, head motion correction, image distortion correction and spatial transformation to the $2 \times 2 \times 2\ mm^3$ Montreal Neurological Institute (MNI) space using one step spline resampling from the original functional images followed by then intensity normalization (*Glasser et al., 2013*). In this study, these minimally preprocessed images were further analyzed using FSL (*Jenkinson et al., 2012*) and AFNI (*Cox, 1996*). Briefly, first the constant, linear and quadratic trend were removed from these functional images. Next, several nuisance signals were regressed from the time course of each voxel using multiple linear regression, including cerebrospinal fluid signal, white matter signal, and Friston's 24 head motion parameters. Then, temporal band-pass filtering (0.01-0.1 Hz) was performed to reduce the influence of low-frequency drift and the high-frequency physiological noise. Finally, 3D spatial smoothing is applied to each volume of the fMRI data using a Gaussian kernel with Full-width at Half Maximum (FWHM) equaling to 4 mm. The first 50 volumes were discarded to suppress equilibration effects and participants without the full 1200 time points in four resting-state runs were also removed from the following analysis. Any data affected by head motion (mean framewise displacement larger than 0.3 mm) were excluded using the protocol of *Power et al. (2014)*. The resulting time courses were used for the construction and analysis of the brain network. We also compared the findings based on the pipeline described above and elsewhere (*Cheng et al., 2016*) with alternative pipelines for data preprocessing including the HCP minimal preprocessing pipeline, and our pipeline but with global signal removal. More details on the effect of the preprocessing pipelines are shown in the Supplementary Material (*Figure 1—figure supplements 1*).

## Construction of the whole-brain functional network

After preprocessing, the whole brain (gray matter) was parcellated into 94 regions of interest (ROI) according to the automated anatomical labeling (AAL2 ) atlas (*Rolls et al., 2015*) (47 regions in each hemisphere), and the time series were extracted in each ROI by averaging the signals of all voxels within that region. The names of the ROIs and their corresponding abbreviations are listed in *Supplementary file 1*. (This is a standard parcellation scheme, which provides a usable number of different divisions for statistical purposes when differences between regions are being investigated, and includes a useful parcellation of the orbitofrontal cortex.). For comparison, we also provide the results (*Figure 1—figure supplement 2*) based on another atlas (*Shen et al., 2013*) which has also been validated in resting state fMRI studies (*Finn et al., 2015*; *Rosenberg et al., 2016*), although its parcellation of the orbitofrontal cortex is less good than the AAL2 atlas. The Pearson cross-correlations between all pairs of regional BOLD signals were calculated for each subject followed by z-transformation to improve normality, and the whole-brain functional connectivity network (94 $\times$ 94 region-based network with 4371 links) was constructed. Finally, the mean functional connectivity across two scans (each scan containing left to right and right to left phase encoding directions) was used for the following analysis to provide a more reliable estimation of functional connectivity. The correlation of the functional connectivities was high between the two different scans as shown in the Supplementary Material (*Figure 1—figure supplement 3*).

## Categorization of participants into groups

The following factors were considered in choosing the measures to define the groups. The main purpose of the current study was to investigate whether there were differences in functional connectivity that were related to smoking or drinking. Accordingly, in the initial analysis, we categorized the participants for the smoking analysis into two groups, one which used significant amounts of cigarettes, and the other using no or low amounts of cigarettes. The same was performed for the division into two groups with respect to the amount of alcohol drunk. In making the cutoff, we ensured that there were reasonable numbers of participants in the high and the low amount groups. After the initial

**Table 2** The demographic characteristics of participants.

| Basic information | | | | | | | |
|---|---|---|---|---|---|---|---|
| Age (year) | Gender (Male/Female) | Handedness | Race (White/Others) | Education (years) | BMI | Head motion | BPDiastolic |
| 28.781±3.696 | 368/463 | 65.842±43.911 | 617/214 | 14.917±1.798 | 26.513±5.238 | 0.352±0.135 | 76.641±10.512 |
| BPSystolic | Total gray matter volume | | DDisc_AUC_200 | DDisc_AUC_40K | Total white matter volume | | |
| 123.702±14.013 | 684867.9±65398.2 | | 0.254±0.201 | 0.499±0.287 | 444066.8±55413.1 | | |

| Smoking information | | | | | | | |
|---|---|---|---|---|---|---|---|
| SSAGA_TB_Smoking_History | SSAGA_TB_Yrs_Smoked | SSAGA_TB_Yrs_Smoked | SSAGA_FTND_Score | SSAGA_HSI_Score | SSAGA_TB_Age_1 st_Cig | SSAGA_TB_Reg_CPD | SSAGA_TB_Hvy_CPD |
| 461/370 | 13.834±4.006 | 13.83±4.01 | 2.063±1.871 | 1.571±1.369 | 16.337±2.303 | 10.098±6.163 | 12.439±7.253 |

| Drinking information | | | | | | | |
|---|---|---|---|---|---|---|---|
| SSAGA_Alc_12_Drinks_Per_Day | SSAGA_Alc_12_Frq | SSAGA_Alc_12_Frq_Drk | SSAGA_Alc_12_Max_Drinks | SSAGA_Alc_Hvy_Frq | SSAGA_Alc_D4_Dp_Sx | SSAGA_Alc_D4_Ab_Dx | SSAGA_Alc_Hvy_Drinks_Per_Day |
| 2.271±1.580 | 4.351±1.530 | 3.114±0.863 | 2.830±1.817 | 3.438±1.777 | 0.518±0.799 | 1.593±1.422 | 3.367±1.745 |

| Marijuana Use information | | | |
|---|---|---|---|
| SSAGA_Mj_Use (No/Yes) | SSAGA_Mj_Ab_Dep (No/Yes) | SSAGA_Mj_Age_1 st_Use | SSAGA_Mj_Times_Used |
| 379/452 | 750/81 | 2.612 ± 0.935 | 1.398 ± 1.693 |

| The comparison of demographic characteristics of different sub-groups | | | | | |
|---|---|---|---|---|---|
| | Non-smoker | Regular Smoker | Statistic*/p value | Low Drinker | High Drinker | Statistic*/p value |
| Number of participants | 417 | 203 | / | 311 | 470 | / |
| Age | 28.50 ± 3.70 | 29.62 ± 3.58 | −3.57/0.0004 | 29.28 ± 3.63 | 28.49 ± 3.70 | 2.94/0.0034 |
| Gender (male/female) | 162/255 | 105/98 | 9.23/0.0024 | 97/214 | 249/221 | 36.01 / < 0.0001 |
| Handedness | 67.42 ± 41.64 | 64.19 ± 45.92 | 0.88/0.381 | 67.77 ± 43.86 | 64.34 ± 44.26 | 1.06/0.288 |
| Race (white/others) | 301/116 | 163/40 | 4.77/0.029 | 219/92 | 369/101 | 6.59/0.010 |
| SSAGA_Educ | 15.23 ± 1.66 | 14.15 ± 1.91 | 7.21 / < 0.0001 | 15.17 ± 1.74 | 14.79 ± 1.83 | 2.91/0.004 |
| BMI | 26.65 ± 5.46 | 26.57 ± 4.73 | 0.19/0.852 | 26.19 ± 5.54 | 26.77 ± 4.92 | −1.52/0.129 |
| Head Motion | 0.087 ± 0.031 | 0.095 ± 0.038 | −2.63/0.009 | 0.085 ± 0.032 | 0.090 ± 0.033 | −1.09/0.057 |
| BPDiastolic | 76.42 ± 10.20 | 77.14 ± 10.42 | −0.82/0.410 | 75.45 ± 10.62 | 77.61 ± 10.15 | −2.86/0.004 |
| BPSystolic | 122.8 ± 13.88 | 125.5 ± 13.86 | −2.26/0.024 | 121.9 ± 13.34 | 125.2 ± 13.99 | −3.33/0.0009 |
| DDisc_AUC_200 | 0.264 ± 0.199 | 0.202 ± 0.166 | 3.83/0.0001 | 0.274 ± 0.210 | 0.232 ± 0.186 | 2.90/0.0039 |
| DDisc_AUC_40K | 0.510 ± 0.286 | 0.438 ± 0.278 | 2.95/0.0033 | 0.516 ± 0.291 | 0.480 ± 0.280 | 1.75/0.081 |
| SSAGA_TB_Smoking_History (never/used) | 417 | 203 | / | 113/98 | 251/219 | 21.91 / < 0.0001 |
| SSAGA_TB_Yrs_Smoked | / | 13.87 ± 3.98 | / | 13.20 ± 4.70 | 14.15 ± 3.61 | −1.56/0.119 |
| SSAGA_Alc_12_Drinks_Per_Day | 1.978 ± 1.478 | 2.842 ± 1.756 | −6.41 / < 0.0001 | 0.823 ± 0.382 | 3.223 ± 1.320 | −31.2 / < 0.0001 |
| SSAGA_Alc_12_Frq | 4.717 ± 1.356 | 3.980 ± 1.668 | 5.88 / < 0.0001 | 5.03 ± 1.29 | 3.91 ± 1.51 | 10.73 / < 0.0001 |
| SSAGA_Mj_Use (never/used) | 256/161 | 29/74 | 122.0 / < 0.0001 | 173/138 | 162/308 | 34.21 / < 0.0001 |
| SSAGA_Mj_Times_Used | 0.779 ± 1.273 | 2.783 ± 1.800 | −15.97 / < 0.0001 | 1.016 ± 1.458 | 1.777 ± 1.791 | −6.245 / < 0.0001 |

Values are n or mean ± SD. The fractions provided in the some rows show the numbers of individuals who did not had the property shown in the column/ the number who did. *: A group difference (independent samples t test or $\chi^2$ test). BPDiastolic: Blood Pressure – Systolic; BPSystolic: Blood Pressure – Diastolic; DDisc_AUC_200: Delay Discounting: Area Under the Curve (AUC) for Discounting of $200; DDisc_AUC_40K: Delay Discounting: Area Under the Curve for Discounting of $40,000; PMAT24_A_CR: Penn Progressive Matrices: Number of Correct Responses; PMAT24_A_RTCR: Penn Progressive Matrices: Median Reaction Time for Correct Responses; Flanker_AgeAdj: NIH Toolbox Flanker Inhibitory Control and Attention Test: Age-Adjusted Scale Score; ListSort_AgeAdj: NIH Toolbox List Sorting Working Memory Test: Age-Adjusted Scale Score; PicSeq_AgeAdj: NIH Toolbox Picture Sequence Memory

Test: Age-Adjusted Scale Score; SCPT_TP: Short Penn Continuous Performance Test: True Positives = Sum of CPN_TP and CPL_TP; SSAGA_TB_Smoking_History: Smoking history: never smoked (0), experimented 1–19 times (1), experimented 20–99 times (2), regular smoker (3); SSAGA_TB_Yrs_Smoked: Years smoked (1–5 years = 5; 6–10 = 10; 11–15 = 15; 16+ = 18); SSAGA_FTND_Score: Fagerstrom FTND (test for nicotine dependence) score: 4 + indicative of dependence;>6 recoded as 6); SSAGA_HSI_Score: Fagerstrom HSI Score: HSI measure of tobacco dependence; SSAGA_TB_Age_1 st_Cig: For regular smokers, age first smoked a cigarette (even a puff); SSAGA_TB_Reg_CPD: Cigarettes per day when smoking regularly; SSAGA_TB_Hvy_CPD: Cigarettes per day during heaviest period; SSAGA_Alc_12_Drinks_Per_Day: Drinks consumed per drinking day in past 12 months: 0, 1, 2, 3, 4, 5–6 = 5, 7+ = 6; SSAGA_Alc_12_Frq: Frequency of any alcohol use in past 12 months; SSAGA_Alc_12_Frq_Drk: Frequency drunk in past 12 months; SSAGA_Alc_12_Max_Drinks: Max drinks in a single day in past 12 months; SSAGA_Alc_Hvy_Frq: Frequency of any alcohol use, heaviest 12 month period; SSAGA_Alc_D4_Dp_Sx: Number of DSM4 Alcohol Dependence Criteria Endorsed; SSAGA_Alc_D4_Ab_Dx: DSM4 Alcohol Abuse Criteria Met; SSAGA_Alc_Hvy_Drinks_Per_Day: Drinks per day in heaviest 12 month period; SSAGA_Mj_Use: Ever used marijuana: no = 0; yes = 1; SSAGA_Mj_Ab_Dep: Participant meets DSM criteria for Marijuana Dependence; SSAGA_Mj_Age_1 st_Use: Age at first marijuana use:<=14 = 1; 15–17 = 2; 18–20 = 3;>=21 = 4; SSAGA_Mj_Times_Used: Times used marijuana: never used = 0; 1–5 = 1; 6–10 = 2; 11–25 = 3; 26–50 = 3; 51–100 = 3; 101–999 = 4; 1000–2000 = 5; >2000 = 5. For more details for each term, it is available on the website: https://wiki.humanconnectome.org/display/PublicData/HCP+Data+Dictionary+Public-+500+Subject+Release

DOI: https://doi.org/10.7554/eLife.40765.019

analysis, we wished to investigate the effect of the different amounts of use of cigarettes or alcohol, and therefore used measures of the different amounts of drinking of each participant, and the number of times each participant had smoked, as described in the results section (*Figures 5* and *6*).

Accordingly, to investigate smoking-associated functional connectivity, we divided all participants into three groups according to their smoking history (SSAGA_TB_Smoking_History), that is non-smokers (417 participants), occasional smokers (161 participants) and regular smokers (203 participants). The non-smoker group contained participants who had never smoked during their lifetime. The occasional smoker group contained participants who experimented with cigarettes at least once in their lifetime but never became a regular smoker. The regular smoker group included those who smoked at least 100 cigarettes in his or her lifetime and smoked at least one day per week when they were smoking regularly. (A score of 3 indicates a regular smoker; of 1 and 2 indicates an occasional smoker, and 0 a non-smoker.) In the main analyses described in this paper to investigate whether there was an effect of smoking, two of these groups were used, the non-smoker and regular smoker groups. The third group was helpful in further analyses to investigate effects of the amount of smoking (*Figure 5*).

To investigate drinking-associated functional connectivities, we divided all participants into two groups. A Low Amount (LA) group with 311 participants drank one or fewer drinks per drinking day in the past 12 months. A High Amount (HA) group with 470 participants drank two or more drinks per drinking day in the past 12 months. (The descriptor was SSAGA_Alc_12_Drinks_Per_Day, and this division into two groups was made to help comparison with the initial analysis for smokers, which also used two groups.) It should be noted that the 'amount' of drinking in the current study refers to the amount of drinks per drinking day (and not for example to the total number of drinks in the past 12 months). The rationale for the use of this measure is provided in the Discussion section, and a complementary analysis showed that there was a good correlation between the association pattern of the functional connectivities when the measures were the 'total drinks' and the 'drinks per drinking day' (r = 0.72, p<1 $\times$ 10$^{-10}$). In order to compare the results for smoking and drinking, only participants with information about both drinking and smoking behavior were included. The demographic characteristics of the participants in each group are summarized in *Table 2*.

## Statistical analysis

In order to investigate the relationship between functional connectivity (FC) and smoking, two sample two-tailed t tests were used to test whether smoking was associated with functional connectivity after removing the confounding effects of age, gender, years of education, race, handedness, head motion (mean framewise displacement, FD), BMI, blood pressure (diastolic and systolic), total gray matter volume, total white matter volume, drinking, and marijuana use. Considering the purpose of this study described above and the sample size in each sub-group, we used a two sample t-test rather than a 2 $\times$ 2 ANOVA. Then a Storey's FDR procedure was used to correct for multiple comparisons (*Storey, 2002*; *Storey and Tibshirani, 2003*; *Erlikhman and Caplovitz, 2017*). Storey's FDR is a modification of FDR, also called positive False Discovery Rate (pFDR) by conditioning on

one false positive finding having occurred. The Storey's FDR method was implemented by the Matlab function mafdr.m with default parameters. In the present study FDR correction for the functional connectivity between any pair of AAL2 regions was used, and the results for smoking are presented based on this statistical test with FDR $p<0.005$. We then performed the same statistical analysis on the high drinking and low drinking groups with the FDR correction at $p<0.05$. The effects of any use of the other substance was always removed by regression when we performed the analysis on the use of one of the substances. For the effect of the most recent drink/cigarette use, a correlation analysis confirmed that the association pattern was highly consistent in the cases with and without the covariates of the total drinks and total times smoked in the past 7 days ($r = 0.957$ for smoking and $r = 0.927$ for drinking). This provides evidence that it is whether one is a smoker or drinker that is relevant to the results described in this paper, and not just the most recent drug use.

## Correlation of functional connectivity differences for substances with extent of use and a measure of impulsivity

We explored the relationship between the functional connectivity links identified above as different in substance use groups with behavioral measures (i.e. drinks per drinking day in the past 12 months and the amount of smoking). Specifically, a partial correlation analysis was used to measure the correlation between the identified functional connectivity and these behavioral measures with removal of the confounding variables of age, gender, years of education, race, handedness, head motion (mean FD), BMI, blood pressure (diastolic and systolic), total gray matter volume, total white matter volume and use of the other substance.

We also investigated the relationship between the functional connectivity links identified above as different in substance use groups and behavioral scores related to self-regulation and impulsivity. We performed a partial correlation of the delay discounting score with the amount of drinking per drinking day and the amount of smoking respectively. The score of self-regulation and impulsivity was based on the delay discounting task which measures the undervaluing of rewards delayed in time. We used an area-under-the-curve discounting measure (AUC) that provides a valid and reliable index of how steeply an individual discounts delayed rewards (*Myerson et al., 2001*). Specifically, we calculated the canonical correlation between the area under the curve for discounting of $200 and $40,000 (DDisc_AUC_200 and DDisc_AUC_40K) with the strength of associated functional connectivities for drinking and smoking respectively, after removing the effects of age, gender, years of education, race, handedness, head motion (mean FD), BMI, blood pressure (diastolic and systolic) total gray matter volume, total white matter volume and use of the other two substances.

## Cross-validation with the IMAGEN dataset

We used another separate large longitudinal fMRI dataset (the IMAGEN dataset (*Schumann et al., 2010*)) which included 1176 participants to test whether the findings based on the HCP dataset could be cross-validated with an independent dataset, which would considerably strengthen the findings. This dataset also enabled examination of whether functional connectivity at age 14 when participants neither smoke nor drank was associated with who would become drinkers or smokers at age 19. This potentially thus provides a way to investigate whether differences between individuals that were not caused by smoking and drinking (as there was little smoking and drinking when the participants were 14) might lead to smoking or drinking by the time that the participants were 19. The IMAGEN dataset also enabled investigation of gender differences in the behaviors that may be predictable from brain functional connectivity in this important period of development. The details of demographic characteristics of participants from the IMAGEN dataset are presented in *Supplementary file 2*.

## Participants

IMAGEN adopted a longitudinal design to collect genetic, neuroimaging, environmental, and behavioural data in the UK and Europe, starting at age 14 years old with a sample 2087 healthy adolescents, and followed at ages 16 and 19 years. The neuroimaging data were acquired at ages 14 and 19 years (*Schumann et al., 2010*). The demographic characteristics of participants who included resting state fMRI scans are summarized in *Supplementary file 2*.

*IMAGEN 19*: For the dataset at age 19, by using a similar criterion to define the sub-groups as in the HCP database, a Low Amount (LA) drinking group with 378 participants who drank two or fewer drinks per drinking day, and a High Amount (HA) drinking group with 566 participants who drank three or more drinks per drinking day, were identified.

The small difference in the different definitions for the drinking groups for the HCP and IMAGEN samples is because they used different questionnaires. We used the European School Survey Project on Alcohol and Other Drugs (ESPAD) to assess the lifetime drinking and smoking in the IMAGEN dataset. The measure of drinking in the IMAGEN dataset is as follows:

"How many drinks containing alcohol do you have on a typical day when you are drinking?

1: 1 or 2
2: 3 or 4
3: 5 or 6
4: 7 to 9
5: 10 or more'

For comparison, the measure of drinking in the HCP dataset is:

'Drinks consumed per drinking day in past 12 months: 0, 1, 2, 3, 4, 5–6 = 5, 7+ = 6.'

In fact, the control group for the HCP and IMAGEN datasets were similar, in that they had not drunk alcohol.

For the smoking dataset, a smoking group (regular smoker) was defined as those who smoked at least 100 cigarettes in his or her lifetime and smoked at least one day per week when they were smoking regularly (180 participants), and a non-smoking group (who had never smoked, 295 participants).

*IMAGEN 14*: Since almost all participants did not take any cigarettes and alcohol at age 14, it was not suitable to assess the difference between smoking and non-smoking group (or low drinking and high drinking group). However, this longitudinal data enabled us to investigate possible causal associations between substance use behaviors and functional connectivities. Therefore, for the dataset at age 14, to investigate possible causal associations between drinking and functional connectivities, we focused only on the subjects who drank two or fewer drinks per drinking day. (This was the great majority of participants.) We also divided all participants into two groups according to their drinking behavior at age 14 and 19. Specifically, a Future Low Amount (FLA) group with 86 participants drank with two or fewer drinks per drinking day at both age 14 and 19. A Future High Amount (FHA) group with 91 participants drank with three or more drinks per drinking day at age 19. It should be noted that both of these groups drank two or fewer drinks per drinking day at age 14.

Similarly to the above, for the analysis of smoking, we still focused on the participants who never smoked cigarettes at age 14: specifically, on a future non-smoking group with 56 participants who never smoked at both age 14 and 19; and a future regular smoking group with 19 participants who experienced smoking at age 19.

Then a two sample two-tailed t-test was used to assess the difference between the (F)LA group and (F)HA group and the smoking and non-smoking groups in the datasets at both the ages of 14 and 19 after removing the effect of gender, BMI, head motion (mean FD) and the use of the another substances use.

We also investigate the impulsivity of participants at age 14 by using the IMAGEN dataset. The impulsivity measure is based on the Substance Use Risk Profile Scale (*Woicik et al., 2009*) by using the sum of the following question scores:

1. I often don't think things through before I speak.
2. I often involve myself in situations that I later regret being involved in.
3. I usually act without stopping to think.
4. Generally, I am an impulsive person.
5. I feel I have to be manipulative to get what I want.

## Drinking and smoking behavior

We used the European School Survey Project on Alcohol and Other Drugs (ESPAD) at ages 14 and 19 years to assess life time drinking and smoking. The primary question of interest was regarding lifetime alcohol and cigarette use and the questionnaire were as follows (*Schumann et al., 2010*):

1. On how many occasions during your lifetime have you smoked cigarettes?

```
0:  0
1:  1–2
2:  3–5
3:  6–9
4:  10–19
5:  20–39
6:  40 or mor
```

2. How frequently have you smoked cigarettes during the last 30 days?

```
  6:  more than 20 cigarettes per day
999:  non smoker
  0:  not at all
  1:  less than one cigarette per week
  2:  less than one cigarette per day
  3:  1–5 cigarettes per day
  4:  6–10 cigarettes per day
  5:  11–20 cigarettes per day
```

3. How many drinks containing alcohol do you have on a typical day when you are drinking?

```
1:  1 or 2
2:  3 or 4
3:  5 or 6
4:  7 to 9
5:  10 or more
```

4. On how many occasions over the last 12 months have you had any alcoholic beverage to drink?

```
0:  0
1:  1–2
2:  3–5
3:  6–9
4:  10–19
5:  20–39
6:  40 or more
```

## Imaging data preprocessing for the IMAGEN dataset

Data preprocessing was performed using DPARSF (*Chao-Gan and Yu-Feng, 2010*) (http:// restfmri. net) which is a toolbox based on the SPM8 software package. The first 10 EPI scans were discarded to suppress equilibration effects. The remaining scans of each subject underwent slice timing correction by sinc interpolating volume slices, motion correction for volume to volume displacement, spatial normalization to standard Montreal Neurological Institute (MNI) space using affine transformation and nonlinear deformation with a voxel size of $3 \times 3 \times 3\mathrm{mm}^3$, followed by spatial smoothing (6 mm Full Width Half Maximum FWHM). To remove the sources of spurious correlations present in resting-state BOLD data, all fMRI time-series underwent band-pass temporal filtering (0.01-0.1 Hz), nuisance signal removal from the ventricles, and deep white matter, and regressing out any effects of head motion using Friston's 24 head motion parameters procedure. Finally, subjects with the mean framewise displacement (FD) of head motion large than 0.3 were completely excluded from the analysis as it is likely that such high-level of movement would have had an influence on several volumes (*Power et al., 2014*).

## Methodological notes
### Effect of the pipeline of data preprocessing
A comparison between different pipelines of data preprocessing was performed, as follows.

1)The complete HCP pipeline.

Data pre-processing was carried out using FSL (FMRIB Software Library), FreeSurfer, and the Connectome Workbench software. All the data preprocessing procedures were performed by the human connectome project (HCP) as described in (*Glasser et al., 2013*). Briefly, the data preprocessing

included correction for spatial and gradient distortions and head motion, intensity normalization and bias field removal, registration to the T1 weighted structural image, transformation to 2 mm Montreal Neurological Institute (MNI) space, and the FIX artefact removal procedure (*Smith et al., 2013*; *Navarro Schröder et al., 2015*).

2)The HCP minimal preprocessing pipeline + other standard preprocessing procedures but without global signal removal.

The HCP minimal preprocessing pipeline was used for the HCP data set. This pipeline includes artifact removal, motion correction and registration to standard space. Then standard preprocessing procedures were applied to the fMRI data: first the linear trend and quadratic term were removed from these functional images; several nuisance signals were regressed from the time course of each voxel using multiple linear regression, including cerebrospinal fluid, white matter and head motion parameters; finally, temporal band-pass filtering (0.09–0.1 Hz) was performed to reduce the influence of low-frequency drift and the high-frequency physiological noise. This pipeline has been widely used in previous studies based on the HCP dataset (*Zalesky et al., 2014*; *Schultz and Cole, 2016*) and our paper also focuses on the results based on this pipeline.

3)The HCP minimal preprocessing pipeline + other standard preprocessing procedures and with global signal removal.

Same as pipeline two but with the global signal removal.

## Acknowledgements

Use of the Human Connectome Project (http://www.humanconnectome.org/) dataset is acknowledged. J Feng is partially supported by the key project of Shanghai Science and Technology Innovation Plan (No. 15JC1400101 and No. 16JC1420402) and the National Natural Science Foundation of China (Grant No. 71661167002 and No. 91630314). The research was also partially supported by the Shanghai AI Platform for Diagnosis and Treatment of Brain Diseases, the Projects of Zhangjiang Hi-Tech District Management Committee, Shanghai (No. 2016–17). The research was also partially supported by Base for Introducing Talents of Discipline to Universities No. B18015. W Cheng is supported by grants from the National Natural Sciences Foundation of China (No.81701773, 11771010), Sponsored by Shanghai Sailing Program (No. 17YF1426200) and the Research Fund for the Doctoral Program of Higher Education of China (No. 2017M610226). W.Cheng is also sponsored by Natural Science Foundation of Shanghai (No. 18ZR1404400). This work also received support from the following sources: the European Union-funded FP6 Integrated Project IMAGEN (Reinforcement-related behaviour in normal brain function and psychopathology) (LSHM-CT- 2007–037286), the Horizon 2020 funded ERC Advanced Grant 'STRATIFY' (Brain network based stratification of reinforcement-related disorders) (695313), ERANID (Understanding the Interplay between Cultural, Biological and Subjective Factors in Drug Use Pathways) (PR-ST-0416–10004), BRIDGET (JPND: BRain Imaging, cognition Dementia and next generation GEnomics) (MR/N027558/1), the FP7 projects IMAGEMEND(602450; IMAging GEnetics for MENtal Disorders) and MATRICS (603016), the Innovative Medicine Initiative Project EU-AIMS (115300–2), the Medical Research Council Grant 'c-VEDA' (Consortium on Vulnerability to Externalizing Disorders and Addictions) (MR/N000390/1), the Swedish Research Council FORMAS, the Medical Research Council, the National Institute for Health Research (NIHR) Biomedical Research Centre at South London and Maudsley NHS Foundation Trust and King's College London, the Bundesministeriumfür Bildung und Forschung (BMBF grants 01GS08152; 01EV0711; eMED SysAlc01Z × 1311A; Forschungsnetz AERIAL), the Deutsche Forschungsgemeinschaft (DFG grants SM 80/7–1, SM 80/7–2, SFB 940/1). Further support was provided by grants from: ANR (project AF12-NEUR0008-01 - WM2NA, and ANR-12-SAMA-0004), the Fondation de France, the Fondation pour la Recherche Médicale, the Mission Interministérielle de Lutte-contre-les-Drogues-et-les-Conduites-Addictives (MILDECA), the Assistance-Publique-Hôpitaux-de-Paris and INSERM (interface grant), Paris Sud University IDEX 2012; the National Institutes of Health, Science Foundation Ireland (16/ERCD/3797), USA (Axon, Testosterone and Mental Health during Adolescence; RO1 MH085772-01A1), and by NIH Consortium grant U54 EB020403, supported by a cross-NIH alliance that funds Big Data to Knowledge Centres of Excellence.

# Additional information

## Funding

| Funder | Grant reference number | Author |
|---|---|---|
| National Natural Science Foundation of China | 81701773 | Wei Cheng |
| Shanghai Sailing Program | 17YF1426200 | Wei Cheng |
| Natural Science Foundation of Shanghai | 18ZR1404400 | Wei Cheng |
| National Natural Science Foundation of China | 11771010 | Wei Cheng |
| National Natural Science Foundation of China | 71661167002 | Jianfeng Feng |
| The Shanghai AI Platform for Diagnosis and Treatment of Brain Diseases | 2016-17 | Jianfeng Feng |
| Base for Introducing Talents of Discipline to Universities | B18015 | Jianfeng Feng |
| National Natural Science Foundation of China | 91630314 | Jianfeng Feng |
| The Key Project of Shanghai Science and Technology Innovation Plan | 15JC1400101 | Jianfeng Feng |
| The Key Project of Shanghai Science and Technology Innovation Plan | 16JC1420402 | Jianfeng Feng |

The funders had no role in study design, data collection and interpretation, or the decision to submit the work for publication.

## Author contributions

Wei Cheng, Conceptualization, Resources, Data curation, Software, Formal analysis, Funding acquisition, Validation, Investigation, Visualization, Methodology, Writing—original draft, Project administration, Writing—review and editing; Edmund T Rolls, Conceptualization, Formal analysis, Supervision, Investigation, Methodology, Writing—original draft, Project administration, Writing—review and editing; Trevor W Robbins, Methodology, Writing—review and editing; Weikang Gong, Zhaowen Liu, Wujun Lv, Hongkai Wen, Software, Methodology; Jingnan Du, Software, Formal analysis; Liang Ma, Software; Erin Burke Quinlan, Hugh Garavan, Eric Artiges, Dimitri Papadopoulos Orfanos, Michael N Smolka, Gunter Schumann, Resources, Data curation; Keith Kendrick, Writing—review and editing; Jianfeng Feng, Conceptualization, Resources, Supervision, Funding acquisition, Investigation, Methodology, Writing—original draft, Project administration, Writing—review and editing

## Author ORCIDs

Wei Cheng (iD) https://orcid.org/0000-0003-1118-1743
Edmund T Rolls (iD) https://orcid.org/0000-0003-3025-1292
Dimitri Papadopoulos Orfanos (iD) http://orcid.org/0000-0002-1242-8990
Keith Kendrick (iD) http://orcid.org/0000-0002-0371-5904
Jianfeng Feng (iD) http://orcid.org/0000-0001-5987-2258

## Ethics

Human subjects: The WU-Minn HCP Consortium obtained full informed consent from all participants, and research procedures and ethical guidelines were followed in accordance with the Washington University Institutional Review Boards (IRB #201204036; Title: 'Mapping the Human Connectome: Structure, Function, and Heritability').

Decision letter and Author response
Decision letter https://doi.org/10.7554/eLife.40765.026
Author response https://doi.org/10.7554/eLife.40765.027

## Additional files

### Supplementary files

• Supplementary file 1. The anatomical regions defined in each hemisphere and their label in the automated anatomical labelling atlas AAL2. Column four provides a set of possible abbreviations for the anatomical descriptions.
DOI: https://doi.org/10.7554/eLife.40765.020

• Supplementary file 2. The demographic characteristics of participants from the IMAGEN dataset.
DOI: https://doi.org/10.7554/eLife.40765.021

• Transparent reporting form
DOI: https://doi.org/10.7554/eLife.40765.022

### Data availability

The dataset used in this study and custom code is available at Dryad.

The following dataset was generated:

| Author(s) | Year | Dataset title | Dataset URL | Database and Identifier |
|---|---|---|---|---|
| Cheng W | 2018 | Data from: Decreased brain connectivity in smoking contrasts with increased connectivity in drinking | https://dx.doi.org/10.5061/dryad.736t01r | Dryad Digital Repository, 10.5061/dryad.736t01r |

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
