## [Decision Letter]

Thank you for submitting your article "Decreased brain connectivity in smoking contrasts with increased connectivity in drinking" for consideration by *eLife*. Your article has been reviewed by three peer reviewers, including Heidi Johansen-Berg as the Reviewing Editor and Reviewer #1, and the evaluation has been overseen by Richard Ivry as the Senior Editor. The following individual involved in review of your submission has agreed to reveal their identity: Dieter Meyerhoff (Reviewer #2).

The reviewers have discussed the reviews with one another and the Reviewing Editor has drafted this decision to help you prepare a revised submission.

Summary:

The authors describe a large sample size, well controlled and cross validated study of similarities and differences between smokers and drinkers. They find a general reduction in brain-wide functional connectivity in smokers and a general increase in brain-wide FC in drinkers. Further, the degree of these changes correlate with a measure of impulsivity. The study seeks to quantify several methodological issues important to neuroimaging (preprocessing pipeline, parcellation schemes, low sample size) and presents a compelling and novel set of findings of contrasting whole brain connectivity patterns associated with exposure to difference substances of abuse.

Essential revisions:

1) Minimization of comorbid results: The minimization of the comorbid results is a missed opportunity. At the least, the smokers who drink should be further characterized and I would encourage the inclusion and interpretation of Supplementary Figure 4 in the main text. As poly drug use and comorbid disorders are the rule not the exception for real world substance use disorder, these results are worthy of more attention. This is relevant in light of the contrasting findings when exposure to each drug is interrogated individually. What is the nature of the interaction in smokers who drink?

2) Methodological concerns: The section "correlation between functional connectivity and the amount of smoking and drinking" appears to suffer from circular logic: Based on smoking group assignment, links that are shown to be different are then correlated with the frequency of smoking? This does not appear to be a valid statistic test as the initial identification of nodes is based on smoking behavior which is then correlated with the results. Can the authors comment?

3) The motivation for the 14 year old to 19 year old IMAGEN data could be more fully articulated. The cross validation with IMAGEN data makes sense, but the authors do not properly introduce a developmental hypothesis. Similarly, the gender specific analysis described in the subsection “Association between the functional connectivity at age 14 and the smoking and drinking at age 19”, are not motivated in the Introduction. Are these post hoc tests?

4) With large sample sizes even small effects can be statistically significant. It is therefore particularly important to comment on effect sizes as well as statistical significance. Some of the effects, particularly the correlations, are very small (r<0.1).

5) The drinking groups are categorised based on number of drinks per drinking day. Although this measure is commonly used in studies of regular drinkers it is less commonly used in community samples that presumably include infrequent drinkers. The authors should better justify their choice and should report whether relationships exist with measures of average/total drinks per week/month. It would also be important to report on what the relationships are between drinks per drinking day and e.g., number of drinking days/total drinks etc.

6) The authors report interesting relationships to behavioural measures of impulsivity. I would like to know how specific these relationships are to impulsivity as opposed other behavioural characteristics. This is particularly important given that these effect sizes are fairly low, despite being statistically significant.

7) Resting FC can fluctuate over time with various factors such as tiredness, time of day, caffeine, mood etc. To what extent have those factors been taken into account? Given the focus of the study it would be particularly relevant to report on when the most recent cigarette/drink was consumed. If smokers/drinkers had recently consumed then that might affect the signal. On the other hand, if the study protocols required abstinence for a period prior to the scan then is it possible that some of effects seen relate to withdrawal/wanting/anxiety? If information about most recent drink/cigarette is not available then it should be commented on as a limitation.

8) There is very little description/characterization of the study participants. How were they recruited? How were they assessed and tested for potential pathologies that could affect the outcome measures? How was comorbid substance use assessed/diagnosed? Any selection based on study targets (drinking/smoking)? This information is necessary when making comparisons to other studies.

9) The manuscript Introduction would greatly benefit from being set in the context of previous work on FC in smokers and drinkers. While the authors cite several smaller studies, they do not really summarize the current knowledge on the topic. And in the Discussion, reference should be made to how this new data fits in with the findings in this previous work.

10) Please clarify why 2 simple t-tests were used with corresponding covariate corrections rather than a 2x2 ANOVA.

11) When discussing 'complimentarity' in the subsection “Comparisons between drinking and smoking” (is that even a word?), the meaning of it and the findings remain rather elusive. Are the authors referring to the possibility that those who are vulnerable to FC alterations associated with drinking may be equally vulnerable to FC alterations (albeit in the other direction and in different brain regions) associated with smoking? Essentially reflecting some vulnerability/resilience to the effects of substance use in general on FC?

12) Although quite catchy in its simplicity, the title of the manuscript should include a hint to the nature of the study participants – these are not individuals diagnosed with a substance use disorder. Some of the findings may be specific to this control population and may be different in pathological cases with full blown substance use disorder diagnoses. This is properly dealt with as a limitation in the Discussion.

13) The Abstract should make reference to the impulsivity measures as well.

14) In the subsection “Statistical Analysis”, give the n for all subgroups or give them in the corresponding Figures 4 and 5.

15) Figure 5A-D could be interpreted to reflect a threshold effect on impulsivity in smokers (only regular smokers show effects). Similarly, among those who drink, any drinking (even 1 drink per day) is associated with greater impulsivity (rather than a high-vs.-low difference implied by the group analyses and regression presented). This brings up the question what happens when you compare no drinking to any drinking? Are the results as strong as low vs. high or will you be power-limited?

[Editors' note: further revisions were requested prior to acceptance, as described below.]

Thank you for resubmitting your work entitled "Decreased brain connectivity in smoking contrasts with increased connectivity in drinking" for further consideration at *eLife*. Your revised article has been favorably evaluated by Richard Ivry (Senior Editor), Heidi Johansen-Berg as Reviewing Editor, and two reviewers.

The manuscript has been improved but there are some remaining issues that need to be addressed before acceptance, as outlined below:

Essential revisions:

1) Brain effects in a community sample of light smokers and/or drinkers may not extrapolate to clinical samples studied previously. This point needs to be more clearly acknowledged in the Discussion. Currently the Discussion mentions in the limitations section that levels of smoking and drinking were within the normal range and so effect sizes may be lower than clinical studies. However, the possibility that different patterns of brain variation may be found within the normal range versus in clinical populations is not mentioned.

2) Regarding original point 15 – we agree with the authors that comparing the small sample of non-drinkers to the much larger sample of those who consumed any alcohol is not a good idea. Given this imbalance, the authors are simply not able to address the issue of a potential threshold effect. This should be acknowledged in the Discussion.

3) The data for individuals who both smoke and drink is given almost no interpretation in the Discussion. The edits to include these results (i.e. Figure 4) in the main manuscript are great. However, the authors should provide some interpretation (however speculative) about why these dual use individuals have a functional connectivity profile that appears to look like that of smokers only. This appears to argue against their idea of "complimentarity." That is, if smoking reduces FC and drinking increases FC *in the same circuits*, why does the combination of smoking and drinking lead to decreases? They don't address this issue in their edited manuscript.

---

## [Author Response]

Essential revisions:1) Minimization of comorbid results: The minimization of the comorbid results is a missed opportunity. At the least, the smokers who drink should be further characterized and I would encourage the inclusion and interpretation of Supplementary Figure 4 in the main text. As poly drug use and comorbid disorders are the rule not the exception for real world substance use disorder, these results are worthy of more attention. This is relevant in light of the contrasting findings when exposure to each drug is interrogated individually. What is the nature of the interaction in smokers who drink?

Thank you for the suggestion, which we have followed: Supplementary Figure 4 has been moved to the main text, and is now Figure 4. To help interpret the results shown in Figure 4, we added a new section “Comparison between smokers, drinkers, and both smokers and drinkers”.

2) Methodological concerns: The section "correlation between functional connectivity and the amount of smoking and drinking" appears to suffer from circular logic: Based on smoking group assignment, links that are shown to be different are then correlated with the frequency of smoking? This does not appear to be a valid statistic test as the initial identification of nodes is based on smoking behavior which is then correlated with the results. Can the authors comment?

First, for the correlation between the mean strength of whole brain functional connectivity and the amount of smoking and drinking, we believe that there is no problem of circular logic. This is because the whole brain functional connectivities are used in this analysis, rather than the links identified in the association analysis.

However, to address the point raised, for the correlation between the mean strength of the identified brain functional connectivity and the amount of smoking and drinking, we have reanalyzed the data without the control group. For smoking, we calculated the partial correlation between the mean strength of the different functional connectivities and the frequency of smoking in the group without non-smokers. We found that the correlation was also significant (r = -0.102, p = 0.031, permutation test) after removing the non-smoking group. In addition, we also performed the same analysis for the individual links by focusing on the group with a history of smoking. The results showed that the correlation value for 269 out of the 273 identified links were negative; and further that 54 out of the 273 identified links were correlated with the number of years smoked (uncorrected p<0.05).

For drinking, we also calculated the partial correlation between the mean strength of the different functional connectivities and the drinks per drinking day in the high drinking group. We found that the correlation tended towards significance (r = 0.069, p = 0.070, permutation test) after removing the low-drinking group.

In addition, we also performed the same analysis for the individual links by focusing on the high drinking group. The results showed that the correlation value of 177 out of 214 identified links were positive, and 38 out of the 214 identified links were correlated with the drinks per drinking day (uncorrected p<0.05).

To address this issue, the following changes have been made to the Results section of the main text of the paper.

"Correlation between the functional connectivity and the amount of smoking and drinking

Figure 5A, B shows that for the smoking group, there was a negative correlation between the mean strength of functional connectivity for both the significantly different links (n=369, r=-0.102, p=0.031, permutation test) and the links for the whole brain (n=830, r=-0.080, p=0.026) with the frequency of smoking across individuals. […] In addition, 177 out of 214 links associated with drinking showed a positive correlation with the drinks per drinking day, and 38 of these 214 identified links were significant (p<0.05 uncorrected)."

3) The motivation for the 14 year old to 19 year old IMAGEN data could be more fully articulated. The cross validation with IMAGEN data makes sense, but the authors do not properly introduce a developmental hypothesis. Similarly, the gender specific analysis described in the subsection “Association between the functional connectivity at age 14 and the smoking and drinking at age 19”, are not motivated in the Introduction. Are these post hoc tests?

We have followed this advice, and added to the Materials and methods section, to make these points more explicit.

"Cross-validation with the IMAGEN dataset

We used another separate large longitudinal fMRI dataset (the IMAGEN dataset (Schumannet al.,2010)) which included 1176 participants to test whether the findings based on the HCP dataset could be cross-validated with an independent dataset, which would considerably strengthen the findings. […] The IMAGEN dataset also enabled investigation of gender differences in the behaviors that may be predictable from brain functional connectivity in this important period of development."

4) With large sample sizes even small effects can be statistically significant. It is therefore particularly important to comment on effect sizes as well as statistical significance. Some of the effects, particularly the correlations, are very small (r<0.1).

First, the HCP dataset was not specifically designed for the study of substance use. It contains numerous behavior measurements, and the use of alcohol and cigarettes are two of them. We cannot expect a high effect size in a sample drawn from the general community, as contrasted with for example a study that selects people with for example heavy smoking and compares then to controls. In more detail, we show that many functional connectivity links are related to drinking and smoking, though each has a relatively low effect size (Figure 1).

Although the effect size for smoking and drinking is relatively low in the HCP dataset, the main findings, such as increased functional connectivity in high drinking group and decreased functional connectivity in smoking group, are robust, in that they were cross-validated in an independent dataset, IMAGEN. Moreover, we do demonstrate with the IMAGEN dataset that the effect size is sufficient to enable prediction from the brain functional connectivities that we identify at age 14 of who will smoke at age 19. This is a good demonstration that the effects described in this paper are meaningful, in that predictions can be made from them about what will happen five years later.

We have followed the advice, and have included the following in the Discussion section of the revised paper to address this point:

"It is also important to stress that all participants were healthy adults, and that the levels of smoking and drinking exhibited were not directly associated with diagnoses of substance dependence, although they may be relevant to these disorders. […] This is good evidence that the effects described in this paper are meaningful, in that the functional connectivity at age 14 can be related to what will happen in terms of smoking and drinking five years later."

5) The drinking groups are categorised based on number of drinks per drinking day. Although this measure is commonly used in studies of regular drinkers it is less commonly used in community samples that presumably include infrequent drinkers. The authors should better justify their choice and should report whether relationships exist with measures of average/total drinks per week/month. It would also be important to report on what the relationships are between drinks per drinking day and e.g., number of drinking days/total drinks etc.

The rationale for the categorization of participants is described in the Materials and methods section, and is considered further in the Discussion as follows: “The amount of alcohol consumed on a drinking day, the ‘amount’ measure, may well be related to how rewarding the consumption of the alcohol is. […] That in turn is consistent with the higher functional connectivity of the medial orbitofrontal cortex found in the High Amount drinkers in this investigation, and with the evidence that the medial orbitofrontal cortex contains a representation of reward value (Rolls, 2014, Rolls, 2017, Rolls, 2018).”

To further address this issue, we also analysed the dataset by using the measure of 'total drinks'. The 'total drinks' is defined as the drinks per drinking day (SSAGA_Alc_12_Drinks_Per_Day) times the frequency of any alcohol use (SSAGA_Alc_12_Frq). We then calculated the partial correlation between the total drinks and functional connectivity after removing the same covariates as in the previous analysis. The results show that the association pattern of the total drinks and the drinks per drinking day are quite similar, as shown by the correlation coefficient (r = 0.721, p<1×10^-10^). To make sure that this point was covered clearly in the paper, we added the following to the Materials and methods section:

"… the amount of drinks per drinking day (and not for example to the total number of drinks in the past 12 months). The rationale for the use of this measure is provided in the Discussion section, and a complementary analysis showed that there was a good correlation between of association pattern of the functional conenctivities when the measures were the 'total drinks' and the 'drinks per drinking day' (r = 0.72, p<1×10^-10^)."

6) The authors report interesting relationships to behavioural measures of impulsivity. I would like to know how specific these relationships are to impulsivity as opposed other behavioural characteristics. This is particularly important given that these effect sizes are fairly low, despite being statistically significant.

To test whether the impulsivity is specifically correlated with the amount of substance use, we calculated the correlation between the amount of substance use and numerous available behavioral measures in the HCP dataset (68 in total), such as emotion, cognition, motor, etc. The details for each item are available in the HCP website

https://wiki.humanconnectome.org/display/PublicData/HCP+Data+Dictionary+Public-+Updated+for+the+1200+Subject+Release. The results are shown in the following table. For the correlation between behaviour assessments and smoking, the top 12 behaviour assessments were all related to impulsivity (Delay Discounting). For the correlation between behaviour assessments and drinking, 6 out of the top 13 behaviour assessments were related to impulsivity. For the canonical correlation between behaviour assessments and FCs associated with smoking, the impulsive measurement (DDisc_AUC_200) ranks 13th. For the canonical correlation between behaviour assessments and FCs associated with drinking, the impulsive measurements (DDisc_AUC_200, and DDisc_AUC_40K) rank 11th and 18th respectively. In general, impulsivity is significantly correlated with both the substance use and the identified links compared with other behaviour assessments.

**Behaviour assessment****Correlation between behaviour assessment and smoking****Correlation between behaviour assessment and drinking****Canonical correlation between behaviour assessment and FCs associated with smoking****Canonical correlation between behaviour assessment and FCs associated with drinking**r valuep valuer valuep valuer valuep valuer valuep value**DDisc_AUC_200****-0.120****5.85E-04****-0.122****6.50E-04****0.643****4.99E-03****0.588****1.59E-03****DDisc_AUC_40K****-0.088****1.15E-02****-0.033****3.59E-01****0.610****1.85E-01****0.579****5.42E-03**PicSeq_Unadj0.0511.42E-010.0205.86E-010.6141.27E-010.5274.53E-01PicSeq_AgeAdj0.0481.69E-010.0195.89E-010.6171.02E-010.5245.21E-01CardSort_Unadj0.0176.20E-010.0551.29E-010.5964.22E-010.5548.93E-02CardSort_AgeAdj0.0166.51E-010.0591.01E-010.5993.62E-010.5596.00E-02Flanker_Unadj-0.0137.20E-01-0.0039.40E-010.6131.38E-010.5392.59E-01Flanker_AgeAdj-0.0049.16E-01-0.0049.16E-010.6131.40E-010.5491.28E-01PMAT24_A_CR-0.0097.89E-010.0009.98E-010.6131.42E-010.5586.52E-02PMAT24_A_SI0.0029.51E-01-0.0313.86E-010.6072.30E-010.5501.27E-01PMAT24_A_RTCR0.0166.49E-010.0098.07E-010.5964.16E-010.5549.24E-02ReadEng_Unadj-0.0461.86E-01-0.0628.54E-020.6574.15E-040.5927.68E-04ReadEng_AgeAdj-0.0432.18E-01-0.0531.43E-010.6621.67E-040.5891.17E-03PicVocab_Unadj-0.0723.96E-02-0.1053.58E-030.6407.54E-030.6102.45E-05PicVocab_AgeAdj-0.0685.10E-02-0.1053.48E-030.6444.01E-030.6141.01E-05ProcSpeed_Unadj0.0392.61E-010.0156.86E-010.6003.30E-010.5323.63E-01ProcSpeed_AgeAdj0.0422.24E-010.0137.21E-010.6032.89E-010.5323.65E-01DDisc_SV_1mo_200-0.1406.08E-05-0.0264.70E-010.6264.23E-020.5586.18E-02DDisc_SV_6mo_200-0.1387.23E-05-0.0841.89E-020.6398.58E-030.5672.43E-02DDisc_SV_1yr_200-0.0956.39E-03-0.1005.28E-030.6331.90E-020.6141.17E-05DDisc_SV_3yr_200-0.1331.36E-04-0.1341.83E-040.6492.02E-030.5805.32E-03DDisc_SV_5yr_200-0.1052.52E-03-0.1199.57E-040.6312.37E-020.5711.66E-02DDisc_SV_10yr_200-0.0694.72E-02-0.0851.85E-020.6425.90E-030.5721.43E-02DDisc_SV_1mo_40K-0.0772.80E-020.0088.19E-010.6101.85E-010.5323.75E-01DDisc_SV_6mo_40K-0.0471.81E-010.0049.08E-010.6226.84E-020.5973.63E-04DDisc_SV_1yr_40K-0.0442.13E-010.0274.61E-010.6539.85E-040.5973.38E-04DDisc_SV_3yr_40K-0.0608.82E-02-0.0501.68E-010.6151.21E-010.5862.19E-03DDisc_SV_5yr_40K-0.1013.68E-03-0.0363.13E-010.5944.62E-010.5624.15E-02DDisc_SV_10yr_40K-0.0841.67E-02-0.0402.71E-010.5964.15E-010.5567.67E-02VSPLOT_TC0.0205.71E-010.0097.97E-010.6199.19E-020.5604.95E-02VSPLOT_CRTE0.0068.68E-01-0.0531.42E-010.6082.07E-010.5392.59E-01VSPLOT_OFF-0.0353.18E-01-0.0422.47E-010.6254.67E-020.5605.35E-02SCPT_TP-0.0058.87E-010.0382.96E-010.6821.68E-060.5881.63E-03SCPT_TN-0.0274.48E-01-0.0353.34E-010.6023.04E-010.5333.62E-01SCPT_FP0.0274.48E-010.0353.34E-010.6023.04E-010.5333.62E-01SCPT_FN0.0058.87E-01-0.0382.96E-010.6821.68E-060.5881.63E-03SCPT_TPRT-0.0039.30E-01-0.0666.77E-020.6003.62E-010.4929.38E-01SCPT_SEN-0.0058.87E-010.0382.96E-010.6821.67E-060.5881.63E-03SCPT_SPEC-0.0274.48E-01-0.0353.34E-010.6023.04E-010.5333.63E-01SCPT_LRNR0.0264.60E-01-0.0353.28E-010.7241.61E-120.6515.41E-10IWRD_TOT-0.0049.17E-01-0.0117.55E-010.6245.38E-020.5186.28E-01IWRD_RTC0.0471.83E-010.0628.34E-020.5993.57E-010.5432.09E-01ListSort_Unadj0.0451.98E-01-0.0264.78E-010.6052.46E-010.5107.55E-01ListSort_AgeAdj0.0422.27E-01-0.0264.67E-010.6023.00E-010.5196.13E-01ER40_CR0.0244.87E-01-0.0068.58E-010.6023.08E-010.5441.87E-01ER40_CRT0.0235.18E-01-0.0049.16E-010.6121.61E-010.5833.28E-03ER40ANG-0.0049.16E-010.0225.44E-010.5964.15E-010.5048.39E-01ER40FEAR0.0521.35E-010.0657.24E-020.5599.44E-010.5245.27E-01ER40HAP-0.0382.71E-01-0.0647.63E-020.5826.92E-010.5353.19E-01ER40NOE-0.0412.45E-01-0.0284.37E-010.6293.14E-020.5333.66E-01ER40SAD0.0561.12E-01-0.0531.40E-010.6072.34E-010.5313.98E-01AngAffect_Unadj0.0901.00E-020.0432.38E-010.6491.86E-030.5107.54E-01AngHostil_Unadj0.0254.66E-010.0412.60E-010.6293.24E-020.5441.95E-01AngAggr_Unadj0.1292.12E-040.1199.23E-040.5983.78E-010.5225.53E-01FearAffect_Unadj0.0608.66E-020.0392.74E-010.6101.90E-010.5323.83E-01FearSomat_Unadj0.0704.64E-020.0363.21E-010.6189.77E-020.5786.20E-03Sadness_Unadj0.0333.44E-01-0.0501.67E-010.6302.98E-020.5304.12E-01LifeSatisf_Unadj-0.0675.50E-02-0.0481.85E-010.6245.37E-020.5117.40E-01MeanPurp_Unadj-0.0743.47E-02-0.1015.05E-030.6141.37E-010.5548.60E-02PosAffect_Unadj-0.0294.13E-01-0.0461.98E-010.5944.53E-010.5225.65E-01Friendship_Unadj-0.0323.61E-010.1626.32E-060.6013.26E-010.5441.94E-01Loneliness_Unadj0.0264.52E-01-0.0304.03E-010.6254.88E-020.5422.21E-01PercHostil_Unadj0.0372.87E-01-0.0176.40E-010.6435.17E-030.5586.51E-02PercReject_Unadj0.1121.28E-03-0.0039.28E-010.6254.88E-020.5176.50E-01EmotSupp_Unadj-0.0637.09E-020.0107.76E-010.6032.87E-010.5077.94E-01InstruSupp_Unadj-0.0303.92E-01-0.0117.62E-010.6171.02E-010.4709.94E-01PercStress_Unadj0.0599.25E-020.0264.75E-010.6371.12E-020.5264.88E-01SelfEff_Unadj0.0039.23E-010.0049.13E-010.6293.16E-020.5633.95E-02

We have addressed the point briefly in the new text added to the end of this part of the Results, as follows:

"We further found that these measures of impulsivity had relatively high correlations, with respect to the 68 behavior measures in the HCP, with the smoking and drinking, and their related functional connectivities."

7) Resting FC can fluctuate over time with various factors such as tiredness, time of day, caffeine, mood etc. To what extent have those factors been taken into account? Given the focus of the study it would be particularly relevant to report on when the most recent cigarette/drink was consumed. If smokers/drinkers had recently consumed then that might affect the signal. On the other hand, if the study protocols required abstinence for a period prior to the scan then is it possible that some of effects seen relate to withdrawal/wanting/anxiety? If information about most recent drink/cigarette is not available then it should be commented on as a limitation.

To minimize any effects of confounding factors in the analyses, we included 13 covariates, including age, gender, years of education, race, handedness, head motion, BMI, blood pressure (diastolic and systolic), etc. The HCP dataset does not include information about tiredness, time of day, caffeine, and mood when the participants came to perform the MRI scanning. But the dataset does contain information about the total drinks and total times smoked in the past 7 days. To confirm that our findings are robust in respect to the most recent drink/cigarette use, we re-analyzed the data by adding the total drinks and total times smoked in the past 7 days as two covariates. For smoking, the association pattern of functional connectivity with these two covariates is very consistent with the results described in the main paper (r = 0.957, r = 0). For drinking, the result with these two covariates also is very consistent with the results described in the main paper (r = 0.927, r = 0). So the association pattern between functional connectivity and the substance use are almost exactly the same in the cases with and without the covariates of the most recent drink/cigarette use. The following changes were made in the Materials and methods section to address this issue.

"The effects of any use of the other substance were always removed by regression when we performed the analysis on the use of one of the substances. […] This provides evidence that it is whether one is a smoker or drinker that is relevant to the results described in this paper, and not just the most recent drug use.”

8) There is very little description/characterization of the study participants. How were they recruited? How were they assessed and tested for potential pathologies that could affect the outcome measures? How was comorbid substance use assessed/diagnosed? Any selection based on study targets (drinking/smoking)? This information is necessary when making comparisons to other studies.

The Human Connectome Project consortium is a public shared large-scale neuroimaging dataset which aims to map macroscopic human brain circuits and their relationship to behavior in a population of 1200 healthy adults. This dataset has been widely used in neuroimaging studies, and more than 500 papers have been published based on this valuable data. The full details on the subject recruitment, visits, behavioral testing and the parameters of the scanner have been provided in the HCP website (https://www.humanconnectome.org/) and the papers published by the HCP team.

It should be noted that the HCP dataset was not specifically designed for the study of substance use. All participants are healthy adults. The following sentences were added in the subsection “Participants and data preprocessing” to make this clear, and to address the point:

"…Our sample includes 831 subjects (ages 22–35 years, 463 females) scanned on a 3-T Siemens connectome-Skyra scanner. [...] It should be noted that the HCP dataset was not specifically designed for the study of substance use, but it is useful for investigating the impact of alcohol and cigarette use in a community sample."

9) The manuscript Introduction would greatly benefit from being set in the context of previous work on FC in smokers and drinkers. While the authors cite several smaller studies, they do not really summarize the current knowledge on the topic. And in the Discussion, reference should be made to how this new data fits in with the findings in this previous work.

To address this point, we have added the following to the Introduction:

"Resting state functional connectivity differences involving the prefrontal cortex and insula have also been described in smokers (Biet al., 2017, Fedota and Stein, 2015, Sutherland and Stein, 2018, Yuan et al.,2016)."

We have not returned to these reviews and investigations in the Discussion, because the sample size used in this investigation, and the cross-validation with an independent sample in the IMAGEN dataset, are more powerful than previous investigations.

10) Please clarify why 2 simple t-tests were used with corresponding covariate corrections rather than a 2x2 ANOVA.

First, as described in the Materials and methods, the main purpose of the current study was to investigate whether there are differences in functional connectivity that are related to smoking or drinking. So, the first step of this study was identifying the functional connectivities associated with smoking (and the same for drinking) and making sure that the association result was not affected by use of the other two substances (drinking and marijuana use). However, the 2×2 ANOVA cannot control for the effect of the use of another substance very well since the control variables are binary, (i.e. low drinking and high drinking, rather than the exact drinks per drinking day). In addition, the 2×2 ANOVA was used to test whether there are any differences between any two sub-groups (i.e. smoking vs. controls; drinking vs. controls; both smoking and drinking vs. controls; drinking vs. smoking; smoking vs. both smoking and drinking; drinking vs. both smoking and drinking;). But what we interested in is the drinking-related effect and the smoking-related effect, rather than all possible comparisons shown in a 2×2 ANOVA model. If a 2×2 ANOVA had been used as primary test in this study, then post hoc test would need to be performed to identify the effects in which we are interested, and that may increase the burden of correction for multiple comparisons.

Second, only non-smokers and regular smokers were considered in the smoking analysis (620 subjects in total). So, if the ANOVA had been used in the association analysis, only 620 subjects would have been left for the final analysis. This would have significantly reduced the sample size available for the drinking analysis.

Finally, the sample size of each group for an ANOVA is not balanced. The sample size of the no-smoking and low drinking group is 198; the sample size of the no-smoking and high drinking group is 219; and the sample size of the smoking and high drinking group is 143. But the sample size of the smoking and low drinking group is 60, which is significantly smaller than other groups.

To make this clear, the following sentence was added in the main paper:

"Considering the purpose of this study described above and the balance of sample size in each sub-group, we used a two sample t-test rather than the 2×2 ANOVA. "

11) When discussing 'complimentarity' in the subsection “Comparisons between drinking and smoking” (is that even a word?), the meaning of it and the findings remain rather elusive. Are the authors referring to the possibility that those who are vulnerable to FC alterations associated with drinking may be equally vulnerable to FC alterations (albeit in the other direction and in different brain regions) associated with smoking? Essentially reflecting some vulnerability/resilience to the effects of substance use in general on FC?

The word being used was 'complementarity' (a noun), meaning that two processes are 'complementary' (an adjective). We have added the following to the Results section to make the point more explicit:

"This finding indicates a complementarity between the functional connectivities in drinkers and smokers. […] This provides evidence that a difference in one direction (increase) may relate to smoking, and in an opposite direction (decrease) to drinking."

12) Although quite catchy in its simplicity, the title of the manuscript should include a hint to the nature of the study participants – these are not individuals diagnosed with a substance use disorder. Some of the findings may be specific to this control population and may be different in pathological cases with full blown substance use disorder diagnoses. This is properly dealt with as a limitation in the Discussion.

We have addressed this point by making it very clear in the first sentence of the Abstract that this is an investigation in a general population, as follows:

"In a group of 831 participants from the general population in the Human Connectome Project, smokers exhibited low overall functional connectivity, and more specifically of the lateral orbitofrontal cortex which is associated with non-reward mechanisms, the adjacent inferior frontal gyrus, and the precuneus."

We also discuss it in the Discussion as follows:

"Possible limitations of this investigation are that linear analyses were used. […] This is good evidence that the effects described in this paper are meaningful, in that the functional connectivity at age 14 can be related to what will happen in terms of smoking and drinking five years later."

13) The Abstract should make reference to the impulsivity measures as well.

We have addressed this by adding the following to the Abstract, as suggested. It was not present previously in order to keep the Abstract short.

"Increased impulsivity was found in smokers, associated with decreased functional connectivity of the non-reward-related lateral orbitofrontal cortex; and increased impulsivity was found in high amount drinkers, associated with increased functional connectivity of the reward-related medial orbitofrontal cortex."

14) In the subsection “Statistical Analysis”, give the n for all subgroups or give them in the corresponding Figures 4 and 5.

The number of subjects for all subgroups has been added to the main text as suggested.

15) Figure 5A-D could be interpreted to reflect a threshold effect on impulsivity in smokers (only regular smokers show effects). Similarly, among those who drink, any drinking (even 1 drink per day) is associated with greater impulsivity (rather than a high-vs.-low difference implied by the group analyses and regression presented). This brings up the question what happens when you compare no drinking to any drinking? Are the results as strong as low vs. high or will you be power-limited?

The rationale for the categorization of participants is described in the Materials and methods. For the smoking analysis, only non-smokers and regular smokers were considered in our analysis. But for drinking, it is difficult just to divide all the partcipants into a no-drinking and a drinking group. This was because only 55 participants did not drink any alcohol in past 12 months. If we categorized participants in this way, the sample size is very unbalanced (55:727). Because of the power limitation, we did not investigate that question.

[Editors' note: further revisions were requested prior to acceptance, as described below.]

Essential revisions:1) Brain effects in a community sample of light smokers and/or drinkers may not extrapolate to clinical samples studied previously. This point needs to be more clearly acknowledged in the Discussion. Currently the Discussion mentions in the limitations section that levels of smoking and drinking were within the normal range and so effect sizes may be lower than clinical studies. However, the possibility that different patterns of brain variation may be found within the normal range versus in clinical populations is not mentioned.

We have carefully addressed what has been reported in clinical populations, and have addressed the point that the results described here may not generalize to all populations, by the following new text added to the Discussion:

"Although the current study was based on the general population, the findings are in general consistent with previous studies based on clinical populations. […] However, the results described here were obtained on populations from the general community, and do not necessarily apply to heavy smokers or drinkers."

2) Regarding original point 15 – we agree with the authors that comparing the small sample of non-drinkers to the much larger sample of those who consumed any alcohol is not a good idea. Given this imbalance, the authors are simply not able to address the issue of a potential threshold effect. This should be acknowledged in the Discussion.

We have added the following t the Discussion, to follow this suggestion:

"Another possible limitation of the drinking analysis is that the control group was a low drinking group rather than a non-drinker group. The reason for this choice was that only 55 participants did not drink any alcohol in the past 12 months, which would have resulted in too small a sample size."

3) The data for individuals who both smoke and drink is given almost no interpretation in the Discussion. The edits to include these results (i.e. Figure 4) in the main manuscript are great. However, the authors should provide some interpretation (however speculative) about why these dual use individuals have a functional connectivity profile that appears to look like that of smokers only. This appears to argue against their idea of "complimentarity." That is, if smoking reduces FC and drinking increases FC in the same circuits, why does the combination of smoking and drinking lead to decreases? They don't address this issue in their edited manuscript.

The complementarity between those who either smoke or drink is left, as it describes the findings. To address that finding about what occurs in those who both smoke and drink we have added the following to the Discussion. Thank you for asking this question.

"As shown in Figure 4, most of the links are decreased in the group that both smokes and drinks. One explanation is that the effect size for the comparison between non-smokers and regular smokers is higher than the effect size for the comparison between low-drinkers and high drinkers."